# Consistent Paths Lead to Truth: Self-Rewarding Reinforcement Learning for LLM Reasoning

**Kongcheng Zhang**[1,2]**, Qi Yao**[2]**, Shunyu Liu**[3*]**, Yingjie Wang**[3]**,**
**Baisheng Lai**[2]**, Jieping Ye**[2]**, Mingli Song**[1]**, Dacheng Tao**[3*]
[1]Zhejiang University, [2]Alibaba Cloud Computing,
[3]College of Computing and Data Science, Nanyang Technological University, Singapore
zhangkc@zju.edu.cn, yq223369@alibaba-inc.com, shunyu.liu@ntu.edu.sg,
yingjiewang1201@gmail.com, laibaisheng@gmail.com, yejieping.ye@alibaba-inc.com
brooksong@zju.edu.cn, dacheng.tao@gmail.com

## Abstract

Recent advances of Reinforcement Learning (RL) have highlighted its potential in complex reasoning tasks, yet effective training often relies on external supervision, which limits the broader applicability. In this work, we propose a novel self-rewarding reinforcement learning framework to enhance Large Language Model (LLM) reasoning by leveraging the consistency of intermediate reasoning states across different reasoning trajectories. Our key insight is that correct responses often exhibit consistent trajectory patterns in terms of model likelihood: their intermediate reasoning states tend to converge toward their own final answers (*high consistency*) with minimal deviation toward other candidates (*low volatility*). Inspired by this observation, we introduce CoVo, an intrinsic reward mechanism that integrates **Co**nsistency and **Vo**latility via a robust vector-space aggregation strategy, complemented by a curiosity bonus to promote diverse exploration. CoVo enables LLMs to perform RL in a self-rewarding manner, offering a scalable pathway for learning to reason without external supervision. Extensive experiments on diverse reasoning benchmarks show that CoVo achieves performance comparable to or even surpassing supervised RL. Our code is available at https://github.com/sastpg/CoVo.

## 1 Introduction

Reinforcement Learning (RL) has emerged as a promising paradigm for enhancing the reasoning capabilities of Large Language Models (LLMs), particularly in complex tasks such as mathematical analysis [14, 7, 34, 12, 58] and algorithmic programming [5, 3, 76]. Recently, DeepSeek-R1 [11] and its follow-up works [64, 55, 60, 17, 6, 67] have demonstrated that RL with outcome-based rewards (*i.e.*, those derived from verifiable answers [11, 23] or reward models [29, 32]) can incentivize LLMs to develop advanced reasoning capabilities, leading to the spontaneous emergence of complex "aha moment" behaviors. However, a critical bottleneck persists: existing RL works heavily rely on *human-annotated labels* or *pretrained reward models* to provide supervision signals. These requirements limit scalability, as high-quality labels are costly to obtain, and reward models often suffer from biases or distributional mismatches in open-ended reasoning scenarios [61, 68, 73, 28, 74, 77, 27].

To address this challenge, several recent works have explored the concept of *self-rewarding* [61, 54, 69, 9], where the model estimates the correctness of its own answers without external supervision. For instance, TTRL [77] uses majority voting over sampled responses to reward high-frequency answers,

---

*Corresponding author.

39th Conference on Neural Information Processing Systems (NeurIPS 2025).

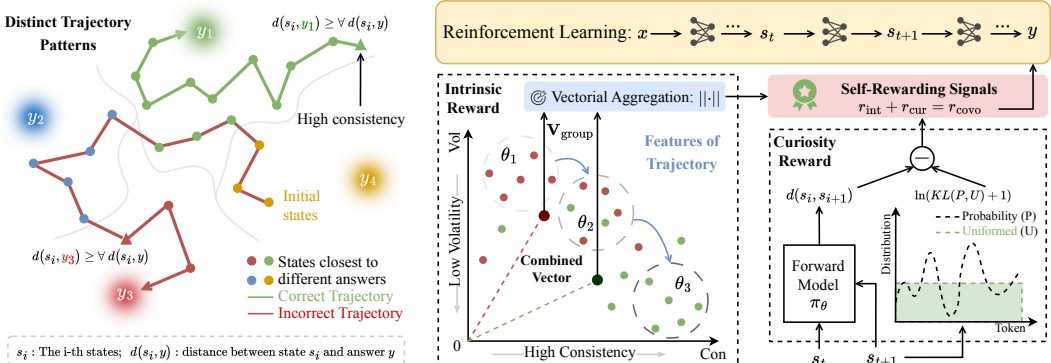

Figure 1: (Left) A conceptual illustration of distinct reasoning patterns: Correct trajectories consistently lead to their own final answer, while incorrect trajectories diverge to different solutions. Different colored nodes along the trajectories denote intermediate reasoning states that are closest to different candidate final answers. (Right) A self-rewarding method that comprises an intrinsic reward to distinguish answer correctness and a curiosity reward to promote diverse exploration.

while EMPO [68] leverages the semantic clusters of answers as a proxy reward signal. However, these methods assign rewards solely based on final answers, overlooking the intermediate reasoning states of reasoning trajectories [47, 49], which is often where errors originate or propagate. As a result, existing self-rewarding techniques for reinforcement learning are highly susceptible to reward hacking, where models may learn to exploit superficial shortcuts that yield unreliable high rewards without genuinely improving reasoning quality (*e.g.*, directly outputting "0" for all math problems can deceive majority voting into rewarding it.). This leads us to a fundamental yet unresolved question:

> *Can intermediate reasoning states be used to derive self-supervised rewards that guide LLMs to generate correct reasoning trajectory without any external supervision?*

In this work, we conceptualize the reasoning trajectory of LLMs as a sequence of intermediate reasoning states with a final answer. To analyze these trajectories, we introduce a distance matrix using model likelihood between each intermediate state and different final answers. Our key finding is that correct and incorrect responses exhibit distinct trajectory patterns in terms of *consistency* (*i.e.*, how many intermediate reasoning states lead to their own final answer) and *volatility* (*i.e.*, how far intermediate reasoning states deviate from their own final answer), as shown in Fig. 1. Unlike prior self-rewarding works that rely only on final answers, this suggests that intermediate reasoning states can provide a rich internal signal for evaluating and guiding reasoning trajectories.

Therefore, we propose a novel self-rewarding RL framework, abbreviated as CoVo, which employs ***Co**nsistency* and ***Vo**latility* as intrinsic rewards to facilitate LLM reasoning. Technically, CoVo introduces a robust vector-space aggregation strategy that geometrically encodes consistency and volatility of reasoning trajectories, enabling effective reward estimation while reducing sensitivity to outlier trajectories. To further encourage diverse exploration, we augment CoVo with a curiosity bonus that incentivizes the model to traverse uncertain yet informative reasoning trajectories based on token probability distribution. Moreover, our theoretical analysis reveals that CoVo in fact treats reasoning trajectories as latent variables within a variational inference objective, thereby grounding rewards on both reasoning paths and final answers. Our core contributions are summarized as:

- We identify an interesting yet underexplored observation that correct responses from LLMs often exhibit *consistent* trajectory patterns, indicating that intermediate reasoning states can serve as a reliable foundation for self-supervised reward construction.

- We propose CoVo, a novel self-rewarding reinforcement learning framework that integrates ***Co**nsistency* and ***Vo**latility* as intrinsic rewards with a curiosity-driven exploration bonus, enabling LLMs to optimize reasoning trajectories without relying on external labels or reward models.

- Extensive experiments on both math-specific and general-purpose benchmarks demonstrate that CoVo significantly improves the reasoning capabilities of LLMs, leading to performances on par with and even sometimes surpassing supervised RL with verifiable answers.

## 2 Preliminaries

In this section, we provide a brief background on LLM reasoning and RL problem formulation.

**Intermediate reasoning states.** We begin by formalizing the concept of intermediate reasoning states and the final answer within a reasoning trajectory. Let $\pi_\theta$ represent a language model parameterized by $\theta$. Given an input prompt $x$ and a reasoning trajectory $\tau$ generated by $\pi_\theta$, we assume the reasoning trajectory can be divided into $T$ discrete steps $t_0, t_1, ..., t_{T-1}$ and a final answer $y$ using predefined rules (*e.g.*, newline characters). We then denote the $i$-th intermediate reasoning state as $s_i = [x, t_0, t_1, ..., t_{i-1}]$. Following the framework established in Zhou et al. [72], we define **distance to final answer** $y$ at state $s_i$ as:

$$d(s_i, y) = -\frac{1}{|y|} \sum_{j=1}^{|y|} \log \pi_\theta(y[j] \mid s_i, y[:j]), \tag{1}$$

where $y$ is the final answer of reasoning trajectory $\tau$, $|y|$ denotes its length. Suppose we sample $N$ reasoning trajectories for an input prompt $x$ and obtain $K$ distinct final answers $\{y_0, y_1, ..., y_{K-1}\}$, we then define **Distance Matrix $\mathbf{D} \in \mathbb{R}^{T \times K}$** for a single trajectory as $\mathbf{D}[i, k] = d(s_i, y_k)$, where $0 \leq i < T, 0 \leq k < K$, and $\mathbf{D}[:, 0]$ denotes the distance to its own final answer. This matrix captures the distances from each intermediate state within one reasoning trajectory to various potential answers generated by other trajectories. It thus provides a fine-grained representation that enables us to analyze how the intermediate reasoning states distinguish patterns of correct responses from erroneous ones. When sampling $N$ reasoning trajectories for a prompt, we will obtain $N$ corresponding distance matrices. In what follows, we focus on prompts where $\pi_\theta$ generates multiple final answers, as cases with all identical sampling answers typically lack meaningful learning signals.

**Reinforcement Learning for LLMs.** Let $\mathcal{V}$ be a finite vocabulary of tokens. Model $\pi_\theta$ receives an input prompt $x \in \mathcal{X}$ and produces a distribution over answers $y \in \mathcal{Y}$, where $\mathcal{X}, \mathcal{Y} \subseteq \mathcal{V}^*$ are the sets of possible input prompts and output sequences, respectively. Following convention in the RLHF literature, we also refer to $\pi_\theta$ as a policy model. Given a reward model (or function) $r$, the policy $\pi_\theta$ is updated by policy gradient methods, such as PPO [40], RLOO [21], GRPO [42], and Reinforce++ [15]. The optimization object is:

$$\mathcal{J}_{\text{RL}}(\theta; x) = \mathbb{E}_{y \sim \pi_\theta(\cdot|x)} \left[ r(x, y) - \lambda \text{KL} \left( \pi_\theta(y|x) \| \pi_{\theta_{\text{ref}}}(y|x) \right) \right], \tag{2}$$

where $\pi_{\text{ref}}$ is the initial policy. We adopt the Reinforce++ [15] algorithm as the backbone in our paper, which combines clipped updates with normalized advantage values through following loss function:

$$\mathcal{L}_{\text{RL}}(\theta) = -\mathbb{E}_t \left[ \min \left( \frac{\pi_\theta(a_i|s_i)}{\pi_{\theta_{\text{old}}}(a_i|s_i)} \hat{A}_i, \text{clip} \left( \frac{\pi_\theta(a_i|s_i)}{\pi_{\theta_{\text{old}}}(a_i|s_i)}, 1 - \epsilon, 1 + \epsilon \right) \hat{A}_i \right) \right], \tag{3}$$

where $\pi_{\theta_{\text{old}}}$ is the previous policy model, $s_i$ is the $i$-th intermediate reasoning states, $a_i \sim \pi_\theta(\cdot|s_i)$ is the selected next token, $\epsilon$ is the clipping coefficient, and the normalized advantage value $\hat{A}_i$ is calculated based on the reward $r$.

## 3 Self-Rewarding Reinforcement Learning

In this section, we aim to elucidate our insight: *How the intermediate states in reasoning trajectories help self-rewarding?* By empirically analyzing distinctive trajectory patterns between correct and incorrect responses, we demonstrate that intermediate reasoning states provide valuable information to distinguish the correctness of responses. After gaining insight into different trajectory patterns, we define two features in the reasoning trajectory and further combine them as self-rewarding signals.

### 3.1 Empirical Analysis: Trajectory Patterns of Different Samples

To investigate whether specific discrepancies exist between correct and incorrect reasoning trajectories, we conduct an empirical study. Specifically, we select two representative reasoning benchmarks MATH [14] and MMLU [13]: the former covers tasks in various mathematical reasoning fields with different difficulty levels; the latter covers general subjects that require world knowledge to reason.

We now examine how intermediate reasoning states in reasoning trajectories converge toward their own final answers. To this end, we sample multiple reasoning trajectories for each input prompt

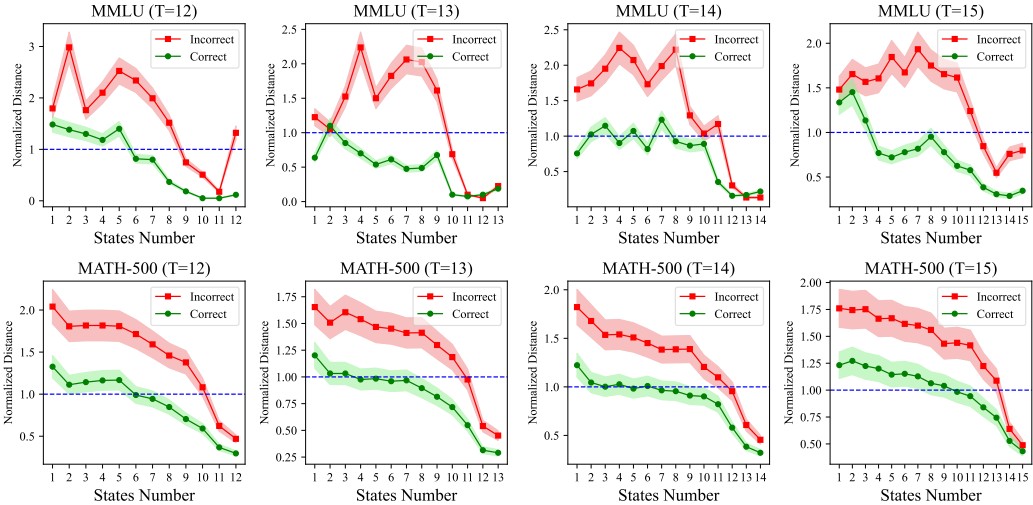

Figure 2: Normalized distance curve of correct and incorrect trajectories with varying state numbers.

and categorize them based on the number of intermediate reasoning states $T$. We then compute the normalized distance $d(s_i, y)/\min_{y' \neq y} d(s_i, y')$ and connect all values as a changing curve, where $y$ is the corresponding final answer of the intermediate reasoning state $s_i$, and $y'$ denotes alternative answers generated by other trajectories. Figure 2 shows the curve comparisons of correct and incorrect reasoning trajectories across varying numbers of intermediate reasoning states, with a value less than 1 indicating an intermediate reasoning state is closest to its own answer.

We find that correct and incorrect responses exhibit distinct trajectory patterns: For correct responses, the reasoning trajectories rapidly "approach" the final answer at early-to-mid states and consistently lead to their own final answer; In contrast, incorrect responses typically exhibit greater fluctuation and delayed convergence, with intermediate reasoning states remaining relatively distant from the final answer for an extended period. This provides strong evidence that the features (consistency and volatility) inside a reasoning trajectory can serve as effective reward signals. We provide results on more reasoning domains to validate this in Section 4.3.

## 3.2 Trajectory Features: Consistency and Volatility

Now we introduce two quantitative features from the distance matrix $\boldsymbol{D}$ that measure the distinct patterns inside a reasoning trajectory $\tau$.

**Consistency**. We first obtain the consistency of a trajectory between the intermediate reasoning states and the final answer. $Con(\tau)$ is the percentage of intermediate states that lead to the corresponding answer $y$ among all different answers:

$$Con(\tau) = \frac{1}{T} \sum_{i=0}^{T-1} \mathbb{I}\left(\mathbf{D}[i,0] = \min_{0 \leq k < K} \mathbf{D}[i,k]\right), \tag{4}$$

where $\mathbb{I}(\cdot)$ is the indicator function and $\mathbf{D}[:,0]$ denotes distance to the answer of trajectory $\tau$ itself.

**Volatility**. We then obtain the volatility of a trajectory between the intermediate states and the final answer. $Vol(\tau)$ measures the extent to which the intermediate reasoning states deviate from their corresponding final answer:

$$Vol(\tau) = \frac{1}{T} \max\left\{ i \in [0, T-1] \mid \mathbf{D}[i,0] \neq \min_{0 \leq k < K} \mathbf{D}[i,k] \right\}. \tag{5}$$

This formula identifies the last state in the reasoning trajectory that deviates from the corresponding answer, and expresses it as a ratio of the total number of intermediate reasoning states.

We briefly discuss the practical significance of these two features in evaluating the quality of reasoning trajectories. Specifically, $Con(\tau)$ captures the prevalence of intermediate reasoning states that favor their own final answer $y$ over other alternatives. Higher consistency can be interpreted as indication

of a more deliberative reasoning process. In contrast, $Vol(\tau)$ reflects how far the reasoning trajectory "strays" toward their answers during the reasoning process evolves. Higher volatility indicates a more erratic and less stable reasoning trajectory prone to flawed or biased decisions.

### 3.3 Self Rewarding in Reinforcement Learning

In Section 3.1 and 3.2, we have demonstrated that correct reasoning trajectories exhibit higher consistency and lower volatility compared to incorrect ones. We now aim to leverage this pattern to develop a self-rewarding reinforcement learning mechanism tailored for reasoning scenarios.

#### 3.3.1 Intrinsic Reward for Self-Evaluation

To formulate a reward signal that distinguishes answer correctness in a batch of reasoning trajectories, we wish to jointly utilize consistency and volatility as the intrinsic reward. During dynamic RL sampling, however, different reasoning trajectories can yield the same final answers, while certain samples may exhibit significant noise in these features due to stochasticity, potentially leading to misjudgment. Inspired by GRPO [42], which uses intra-group relative rewards to estimate baselines, we group trajectories by their final answers and assign intra-group rewards. Specifically, we partition the sampled $N$ reasoning trajectories into groups based on their final answers, resulting in $K$ distinct groups. Let $G$ denote the number of samples in a group that share the same final answer.

A straightforward approach is to linearly aggregate the numerical difference between consistency and volatility within each group. We define the reward function $r_{\text{int}}^{L}$ as the subtraction of consistency and volatility for each trajectory group:

$$r_{\text{int}}^{L} = \frac{1}{G} \cdot \sum_{i=0}^{G-1} \left( Con(\tau_i) - Vol(\tau_i) \right). \tag{6}$$

This linear aggregation directly reflects the intuition that higher consistency and lower volatility indicate a more reliable reasoning trajectory. However, it suffers from sensitivity to extreme outliers, *i.e.*, abnormally low consistency or high volatility can distort the group's overall reward. To mitigate this, we propose a vectorial aggregation that embeds consistency and volatility into a vector space. Under this formulation, each trajectory is represented by a vector composed of these two features:

$$\boldsymbol{v}_i = Con(\tau_i) \cdot \left[ \cos(Vol(\tau_i)), \sin(Vol(\tau_i)) \right], \tag{7}$$

these trajectory-specific vectors are then summed within a group to produce a combined vector:

$$\boldsymbol{V}_{\text{group}} = \sum_{i=0}^{G-1} \boldsymbol{v}_i = \left( \sum_i Con(\tau_i) \cos(Vol(\tau_i)), \sum_i Con(\tau_i) \sin(Vol(\tau_i)) \right). \tag{8}$$

The magnitude of this combined vector serves as the final reward signal $r_{\text{int}}^{V}$ for the group:

$$r_{\text{int}}^{V} = \frac{1}{G} \| \boldsymbol{V}_{\text{group}} \| = \frac{1}{G} \sqrt{ \left( \sum_{i=0}^{G-1} Con(\tau_i) \cos(Vol(\tau_i)) \right)^2 + \left( \sum_{i=0}^{G-1} Con(\tau_i) \sin(Vol(\tau_i)) \right)^2 }. \tag{9}$$

The geometric formulation of $r_{\text{int}}^{V}$ preserves the interplay between consistency and volatility while exhibiting greater robustness to outliers. Moreover, it retains the same monotonic behavior with respect to variations in $Con(\tau)$ and $Vol(\tau)$ as $r_{\text{int}}^{L}$. A formal proof is provided in Appendix B.

#### 3.3.2 Curiosity Reward for Self-Exploration

Our intrinsic reward relies on contrasting diverse reasoning trajectories to identify reliable samples based on their internal consistency and volatility. A recent study [62] reveals that model diversity may decrease as training proceeds, leading to homogeneous results. This poses difficulty for the intrinsic reward to perform meaningful comparisons across distinct answers. To address this, we introduce a curiosity reward to promote a broader exploration of the solution space during RL sampling.

A primary measure of curiosity is the average token probabilities under transitions from intermediate reasoning states, which can be denoted as $d(s_i, s_{i+1})$. This quantifies how the model "anticipates" the subsequent reasoning states. Ideally, the curiosity reward should be assigned to the states where

the model exhibits lower probabilities for exploration. To prevent the curiosity reward from being overly dominant by a few tokens with extremely low probabilities, we introduce a penalty term $p_{\text{KL}}$ based on the KL divergence from a uniform distribution. The curiosity reward is then defined as:

$$r_{\text{cur}} = d(s_i, s_{i+1}) - p_{\text{KL}} = -\frac{1}{|s_{i+1}| - |s_i|} \sum_{j=|s_i|}^{|s_{i+1}|} \log \pi_\theta(s_{i+1}[j] \mid s_{i+1}[:j]) - \ln[KL(P_{i+1}, \mathcal{U}) + 1],$$

(10)

where $\mathcal{U}$ is the uniform distribution, $P_{i+1}$ represents token probability distribution of the state $s_{i+1}$.

Overall, our self-rewarding reinforcement learning framework integrates both self-evaluation reward and self-exploration reward into a total reward $r_{\text{covo}} = r_{\text{int}} + r_{\text{cur}}$. The detailed algorithmic process of reward calculation is provided in Appendix C.

## 4 Experiments

### 4.1 Environmental Setups

**Datasets.** For training, we only adopt instructions from the training set of Open-Reasoner-Zero [17] as our training data, *without relying on any labels*. For evaluation, we employ a variety of datasets that can be categorized into three popular reasoning domains: (1) *Mathematical* reasoning benchmarks include GSM8K [7], MATH-500 [14], Olympiad Bench [12] and AMC-23[1]; (2) *Commonsense* reasoning benchmarks include MMLU-Pro [50] and CommonsenseQA [45]; (3) *Science* reasoning benchmarks include GPQA [39]. These datasets provide essential assessments on the reasoning capabilities of LLMs. More information about our training and evaluation dataset is in Appendix D.

**Models and Baselines.** CoVo is generally applicable to a wide range of LLMs with different backbones. In our experiments, we select three popular open-source LLMs: Llama3.2-3B-Instruct [8], Qwen2.5-3B-Instruct [57], and Qwen2.5-7B-Instruct [57]. By focusing on LLMs with relatively small parameters under 10B, we expect that CoVo enables the smaller model to learn to reason via self-rewarding RL, ultimately achieving results comparable to or exceeding supervised RL in complex reasoning tasks. We compare CoVo against three categories of baselines in our experiments: (1) *In-context learning methods*, including Zero-shot CoT [52], Few-shot CoT, and CoT+SC@4 [46]; (2) *supervised RL methods* that use ground-truth label to compute rule-based reward, including GRPO [42], RLOO [21], Reinforce [66] and Reinforce++ [15]; (3) *Self-rewarding RL methods*, including IRPO [33], EMPO [68] and TTRL [77].

**Evaluation Metric.** We will evaluate both the *Performance* and *Reasoning Diversity* of CoVo in our experiments. For performance evaluation, we employ Pass@1 accuracy under a sampling temperature of 0 across all benchmark datasets as our metric, with correctness determined through comparison between generated outputs and ground truth using Math-Verify [22]. The detailed implementation settings and more evaluation results can be found in Appendix A.

### 4.2 Main Results

**Performance Comparison.** We conduct a comprehensive evaluation of the effectiveness of CoVo across different reasoning domains. Table 1 presents comparative results between our method and state-of-the-art baselines. We find CoVo achieves performance *comparable to or even surpassing supervised RL methods* and is applicable to different model backbones. For example, when applied to Llama3.2-3B-Instruct, CoVo achieves 51.2% accuracy on par with rule-based GRPO (51.8%) on MATH benchmarks. Notably, Qwen2.5-3B-Instruct initially achieves 65.4% accuracy on the MATH benchmark using the few-shot CoT prompting technique. After training by CoVo, its performance improves to 68.2%, which outperforms rule-based supervised RL methods like RLOO and GRPO. We also provide experiments on Qwen2.5-7B-Instruct to demonstrate the effectiveness of CoVo when model parameter scaling in Appendix A.

**Reasoning Boundary.** As presented in Fig. 3, our experiments demonstrate that both TTRL and CoVo improve pass@$k$ scores on the challenging AIME-24 benchmark[2] across all $k$. CoVo outperforms both TTRL and the initial model as $k$ increases, validating its advantage in expanding the reasoning

---

[1]https://huggingface.co/datasets/AI-MO/aimo-validation-amc
[2]https://huggingface.co/datasets/AI-MO/aimo-validation-aime

Table 1: Results of all methods in diverse reasoning domains. *Italics* denotes that this method is training-free, "*" denotes that this method requires sampling multiple outputs for inferences, "†" denotes that this method utilizes self-rewarding without external labels, underline represents the best performance among all baselines, **bold** represents the best performance among all methods.

| Model I: Llama3.2-3B-Instruct (Accuracy ↑) | | | | | | |
|---|---|---|---|---|---|---|
| **Method** | MATH-500 | GSM8K | AMC-23 | Olympiad | MMLU-Pro | GPQA | CommonsenseQA |
| 1. *Zeroshot-CoT* | 47.0 | 75.8 | 17.5 | 15.7 | 34.9 | 28.8 | 71.2 |
| 2. *Fewshot-CoT* | 48.2 | 77.7 | 17.5 | 17.5 | 36.4 | 31.5 | 72.5 |
| 3. *CoT+SC@4 * | 49.6 | 78.9 | 20.0 | 18.8 | 37.1 | 31.9 | 73.6 |
| 4. GRPO | **51.8** | 79.2 | **25.0** | 19.0 | 36.8 | 31.5 | 74.2 |
| 5. RLOO | 51.4 | 79.4 | 22.5 | 19.3 | 36.2 | 31.5 | 73.9 |
| 6. IRPO | 47.2 | 77.5 | 17.5 | 16.3 | 34.7 | 30.1 | 71.8 |
| 7. Reinforce | 51.4 | 79.1 | 20.0 | 19.1 | 36.6 | 31.7 | 73.9 |
| 8. Reinforce++ | 51.6 | **79.6** | 22.5 | 19.4 | 37.1 | 31.9 | 74.1 |
| 9. EMPO † | 51.2 | 79.3 | 22.5 | 19.3 | 37.2 | 31.5 | 73.8 |
| 10. TTRL † | 51.0 | 79.5 | **25.0** | 19.1 | 36.4 | 31.7 | 74.1 |
| **CoVo (Ours) †** | 51.2 | **79.6** | **25.0** | **19.6** | **37.2** | **32.1** | **74.4** |

| Model II: Qwen2.5-3B-Instruct (Accuracy ↑) | | | | | | |
|---|---|---|---|---|---|---|
| **Method** | MATH-500 | GSM8K | AMC-23 | Olympiad | MMLU-Pro | GPQA | CommonsenseQA |
| 1. *Zeroshot-CoT* | 63.6 | 84.8 | 40.0 | 27.9 | 41.7 | 29.7 | 73.6 |
| 2. *Fewshot-CoT* | 65.4 | 86.2 | 37.5 | 28.6 | 43.4 | 30.8 | 75.3 |
| 3. *CoT+SC@4 * | 66.2 | 86.9 | 40.0 | 29.0 | 43.8 | 30.6 | **75.7** |
| 4. GRPO | 67.4 | 88.7 | 45.0 | 29.3 | 43.2 | 31.3 | 75.5 |
| 5. RLOO | 66.8 | **89.1** | 42.5 | 28.6 | 42.6 | 30.8 | 75.6 |
| 6. IRPO | 65.8 | 87.2 | 37.5 | 27.4 | 41.5 | 29.2 | 73.8 |
| 7. Reinforce | 67.0 | 88.4 | 42.5 | 29.5 | 43.2 | 31.3 | 75.8 |
| 8. Reinforce++ | 67.6 | 88.7 | 45.0 | 29.2 | 43.6 | 31.5 | 75.2 |
| 9. EMPO † | 67.4 | 88.5 | 42.5 | 29.3 | **43.9** | 31.3 | 75.3 |
| 10. TTRL † | 67.8 | 88.6 | 45.0 | 29.5 | 43.2 | 31.3 | 75.5 |
| **CoVo (Ours) †** | **68.2** | 88.7 | **47.5** | **29.6** | 43.5 | **31.7** | **75.7** |

diversity. We attribute this enhanced diversity to our reward mechanism that incentivizes novel reasoning paths, as formalized in Eq. (10). Further visualization (Appendix A) indicates that the reasoning paths of our method exhibit a more dispersed distribution, showing that CoVo promotes diverse reasoning strategies rather than collapsing into homogeneous solutions.

**Domain Generalization.** While our training dataset solely consisting of data from the mathematical reasoning domain, CoVo has also shown strong generalization ability across other reasoning domains. Notably, in comparisons between commonsense reasoning and science reasoning tasks, all benchmarks exhibit comparable performance gains relative to other strong baselines, indicating that the reasoning capabilities of CoVo acquired from mathematical training can also effectively transfer to unseen general problem sets.

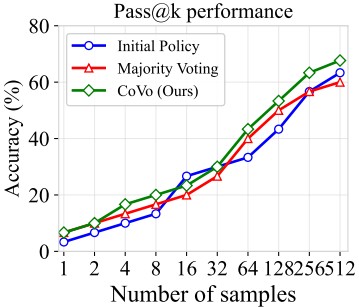

Figure 3: Pass@$k$ curve of Qwen2.5-3B-Instruct model and its counterparts trained with different methods.

### 4.3 Analysis

**Features Discrepancy.** In Section 3, we have conducted qualitative and quantitative analysis on the discrepancies of consistency and volatility across different reasoning trajectories. Now we report the specific statistical data (*i.e.*, the values of consistency and volatility in all responses) on four widely-used reasoning datasets in Table 2, aiming to provide more rigorous empirical support for our insights. To facilitate a more intuitive understanding of the underlying patterns behind the data, we further visualize the statistics in the form of a two-dimensional distribution in Fig. 4. The experimental investigations on the two features reveal clear discrepancies between correct and incorrect responses, which further validate the effectiveness of consistency and volatility as intrinsic indicators to distinguish correct and incorrect reasoning trajectories.

Table 2: The statistical data of consistency and volatility between correct and incorrect responses.

| Feature | Responses | MATH-500 | MMLU | GPQA | CommonsenseQA |
|---------|-----------|----------|------|------|---------------|
| Consistency | Correct | $0.832\pm0.194$ | $0.818\pm0.112$ | $0.787\pm0.126$ | $0.889\pm0.133$ |
| | Incorrect | $0.215\pm0.136$ | $0.236\pm0.087$ | $0.218\pm0.093$ | $0.274\pm0.129$ |
| Volatility | Correct | $0.271\pm0.238$ | $0.310\pm0.255$ | $0.386\pm0.268$ | $0.227\pm0.195$ |
| | Incorrect | $0.821\pm0.117$ | $0.822\pm0.087$ | $0.867\pm0.129$ | $0.761\pm0.156$ |

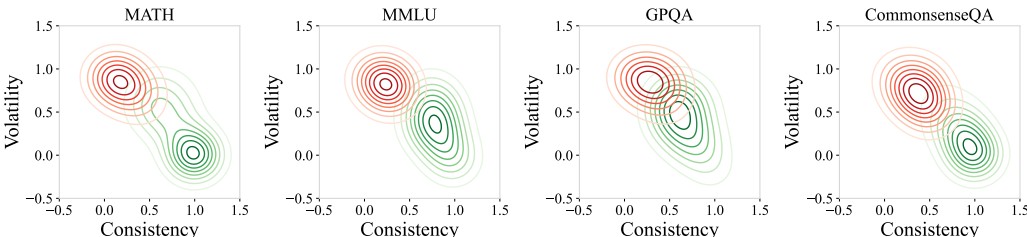

Figure 4: The distribution of consistency and volatility for correct and incorrect responses across diverse domains, where green and red curves denote the distributions for correct responses and incorrect responses, respectively.

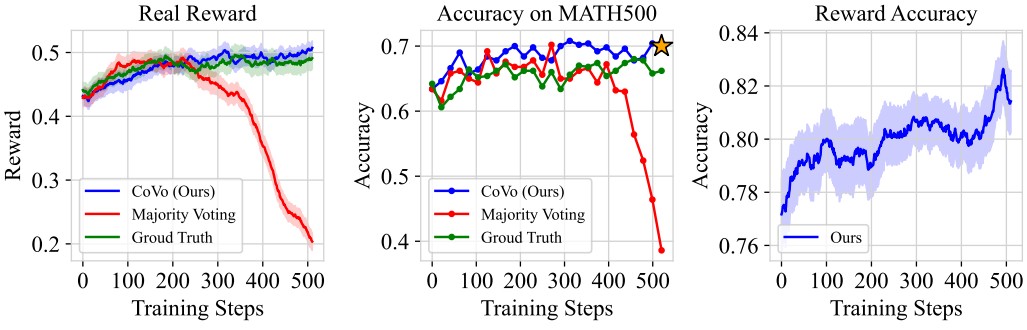

Figure 5: Training dynamics with different types of reward signals using Qwen2.5-3B-Instruct.

**Training Dynamics.** We further present the training variation curves of CoVo on four key metrics during training process. The results in Fig. 5 demonstrate that our reward accuracy remains consistently high and exhibits greater stability against potential reward hacking compared to majority-voting-based reward signals. This stands in contrast to other self-rewarding methods like TTRL and EMPO, which rely on majority voting and answer probabilities for reward assignment.

### 4.4 Ablation Study

Our self-rewarding mechanism integrates two reward signals: an intrinsic reward and a curiosity reward. Note that the intrinsic reward is implemented using two alternative forms, including a linear aggregation $r_{\text{int}}^{L}$ and a vectorial aggregation $r_{\text{int}}^{V}$. To evaluate their contribution to the performance, we conduct ablation studies. Table 3 reports the results of CoVo across different reasoning domains under five different reward configurations. The results show that in almost all reasoning tasks, the combination of intrinsic and curiosity rewards consistently outperforms the individual reward signal, indicating a complementary and positive influence from both intrinsic and curiosity rewards. Moreover, we observe that employing the intrinsic reward $r_{\text{int}}^{V}$ based on vectorial aggregation yields superior accuracy on mathematical and scientific reasoning tasks, which suggests that $r_{\text{int}}^{V}$ is less susceptible to sample noise in scenarios that require substantial arithmetic and analytical reasoning. These findings support our claim in Section 3.3 that $r_{\text{int}}^{V}$ can be more robust than $r_{\text{int}}^{L}$.

## 5 Theoretical Analysis

In this section, we present theoretical foundations on self-rewarding reinforcement learning. To maintain conciseness, the full proof of all propositions are deferred to Appendix B.

Table 3: Ablation study of reward combinations, **bold** represents the best performance.

| Reward | | | Llama3.2-3B-Instruct | | | | | | |
|---|---|---|---|---|---|---|---|---|---|
| $r_{\text{int}}^{V}$ | $r_{\text{int}}^{L}$ | $r_{\text{cur}}$ | MATH-500 | GSM8K | AMC-23 | Olympiad | MMLU-Pro | GPQA | CommonsenseQA |
| ✓ | | ✓ | 51.2 | **79.6** | **25.0** | 19.6 | 37.2 | 32.1 | **74.4** |
| | ✓ | ✓ | 51.0 | 79.2 | **25.0** | 18.8 | **37.8** | 31.5 | 74.2 |
| ✓ | | | **51.6** | 79.5 | 22.5 | **20.6** | 36.2 | **32.4** | 73.8 |
| | ✓ | | 50.8 | 78.3 | 20.0 | 18.4 | 36.9 | 31.5 | 73.3 |
| | | ✓ | 46.2 | 69.5 | 15.0 | 8.1 | 28.7 | 19.6 | 65.5 |

| Reward | | | Qwen2.5-3B-Instruct | | | | | | |
|---|---|---|---|---|---|---|---|---|---|
| $r_{\text{int}}^{V}$ | $r_{\text{int}}^{L}$ | $r_{\text{cur}}$ | MATH-500 | GSM8K | AMC-23 | Olympiad | MMLU-Pro | GPQA | CommonsenseQA |
| ✓ | | ✓ | **68.2** | 88.7 | **47.5** | **29.6** | 43.5 | **31.7** | 75.7 |
| | ✓ | ✓ | 67.8 | **88.9** | 42.5 | 29.0 | **43.8** | 31.0 | **75.9** |
| ✓ | | | 68.0 | 88.4 | 42.5 | 29.3 | 42.9 | 31.3 | 75.1 |
| | ✓ | | 66.4 | 87.9 | 40.0 | 28.3 | 43.1 | 30.6 | 74.7 |
| | | ✓ | 61.8 | 80.6 | 32.5 | 23.9 | 36.8 | 23.4 | 70.4 |

## 5.1 Rethinking Majority Voting in RL Optimization

**Proposition 1** (Model Collapse). *Given an input prompt $x \in \mathcal{X}$, the policy model sample $n$ responses $\{y_1, y_2, \ldots, y_n\} \subseteq \pi_\theta(\cdot|x)$ and denote $C(y) = \sum_{i=1}^{n} \mathbb{I}\{y_i = y\}$. Suppose $y^* \in \mathcal{Y}$ that satisfies $y^* = \arg\max_{y \in \mathcal{Y}} \pi_\theta(y|x)$ and a reward function $r : \mathcal{X} \times \mathcal{Y}$ defined by*

$$r(x, y) = \begin{cases} 1, & \text{if } y = \arg\max_{y' \in \mathcal{Y}} C(y'), \\ 0, & \text{otherwise.} \end{cases} \quad (11)$$

*such that the following hold:*

- *When using gradient ascent to maximize the RLHF objective with respect to the reward function $r$, $\pi_\theta(y^* \mid x)$ converges to 1 as training proceeds.*

- *If $y^*$ is not the ground truth, reward hacking will happen as training proceeds.*

**Remark.** Intuitively, the optimization object of majority-voting-based reward in reinforcement learning is $\max_\theta \mathbb{E}_{y \sim \pi(\cdot|x)} [\log \pi_\theta(y^*|x)]$, which diverges from true answer distribution when model priors misalign with ground truth. This explains why naively optimizing for majority voting causes performance collapse as training proceeds in Fig 5.

## 5.2 Theoretical Foundations of Self-Rewarding Reinforcement Learning

**Proposition 2** (Variational Optimization for Reasoning). *Let $\pi_\theta(y|x)$ denote a language model that generates final answers $y$ conditioned on input prompt $x$, with $s$ representing latent intermediate reasoning states. Suppose the reasoning state is guided by a prior distribution $\pi_{\theta(0)}(s|x)$ and define the reward function $r(s, x, y) \in [0, 1] \propto \log \pi_\theta(y|x \oplus s)$, the following optimizing bound holds:*

$$\log \pi_\theta(y|x) \geq \underbrace{\mathbb{E}_{s, y \sim \pi_\theta(\cdot|x)} [r(s, y, x)]}_{\text{Reward Assignment}} - \underbrace{D_{KL} [\pi_\theta(s|x) \| \pi_{\theta(0)}(s|x)]}_{\text{KL Regularization}}, \quad (12)$$

*where $\oplus$ denotes the concatenation of the input prompt $x$ with the reasoning state $s$.*

**Remark.** This formulation exactly represents the standard optimization objective defined in Eq. (2), illustrating that our self-rewarding RL can be viewed as performing approximate variational inference over latent reasoning trajectories. By leveraging this framework, our reward used for optimization is grounded in the intermediate reasoning states $s$ and final answers $y$, providing a theoretical foundation for consistent reasoning.

## 5.3 Complexity and Convergence of CoVo

**Complexity.** The computational overhead of our self-rewarding mechanism primarily stems from the distance matrices across reasoning trajectories. Let $N$ denote the number of sampled paths per

prompt, $T$ the maximum reasoning steps, and $K$ the number of distinct answer candidates. For each trajectory, computing the distance matrix $\mathbf{D} \in \mathbb{R}^{T \times K}$ requires $\mathcal{O}(TK)$ computations. Aggregating all $N$ trajectories per sample batch incurs $\mathcal{O}(NTK)$ computational complexity.

**Convergence.** To theoretically characterize the optimization behavior, we establish the following proposition built upon the theory in Razin et al. [37].

**Proposition 3** (Convergence). *Let $\mathcal{X}$ denote prompt sets, and $\gamma > 0$ represent the expected increase of real reward $r^*$. For a prompt $x \in \mathcal{X}$, let $y^\gamma$ denote the final answer in a reasoning trajectory with higher consistency and lower volatility. Define $T'$ as the initial time where $\mathbb{E}_{y \sim \pi_\theta(\cdot|x)}\left[r^*(x, y)\right] \geq \mathbb{E}_{y \sim \pi_{\theta(0)}(\cdot|x)}\left[r^*(x, y)\right] + \gamma$. We assume the following conditions hold:*

- *Our reward function ranks $y^\gamma$ highest among all candidates;*

- *Our reward function has a low probability of misclassifying incorrect trajectories;*

- *The KL regularization coefficient is sufficiently small.*

*Then the convergence time for achieving the expected increase in real reward satisfies the upper bound $T' \leq \mathcal{O}(\pi_{\theta(0)}(y^\gamma|x)^{-1})$.*

**Remark.** The inverse dependence on $\pi_{\theta(0)}(y^\gamma|x)$ implies that trajectories initially assigned higher likelihood under the base policy converge faster. This aligns with perspectives in recent studies [62, 10]: RL efficiently amplifies existing reasoning capability rather than instilling entirely new behaviors.

# 6   Related Work

**RL for Reasoning.** Reinforcement Learning [44, 40, 20, 19] has emerged as a prominent paradigm to enhance the capabilities of LLMs. Initial efforts [32, 4] demonstrated the efficacy of RL by applying PPO [40] to align the behavior of the language model with human preference. Subsequent works [18, 42, 25, 65] apply RL to improve the capability of LLMs to solve complex reasoning tasks, focusing on reward model [29, 25, 43] and RL algorithm [59, 63, 42]. Recently, Deepseek-R1 [11] shows that RL with verifiable rewards can incentivize LLMs with advanced reasoning capabilities. Despite its success, existing methods rely heavily on human-annotated labels or external reward models, which constrains their scalability to new scenarios [61]. In contrast, our method mitigates this dependency by deriving reward signals directly from the policy model itself.

**Self-Rewarding LLM.** Self-rewarding LLM aims to generate its own reward signals to guide learning and optimization. Early methods [61, 54, 56] leverage the policy model itself, often through "LLM-as-a-Judge" prompting, to generate both the evaluation criteria and the final rewards. CREAM [51] and SCIR [70] employ ensembles of multiple internal reward models to estimate a more reliable reward signal. PSRLM [69] and CSR [71] further extend self-rewarding mechanisms to intermediate reasoning steps to enable fine-grained reward signals. However, prompting LLMs to generate rewards can be resource-intensive. Consequently, such approaches are typically employed in offline iterative DPO [35]. Although TTRL [77] and EMPO [68] propose methods to compute rewards directly from multiple sampled outputs within an online RL framework, their reliance solely on the distribution of final answers can lead to reward hacking. In contrast, our method incorporates the distinct patterns inherent in reasoning trajectories to get more robust reward signals.

# 7   Conclusion

This work proposes a novel self-rewarding reinforcement learning method to enhance the reasoning capabilities of LLMs without relying on external signals. Our insight stems from distinct patterns observed in reasoning trajectories: Correct trajectories typically exhibit high consistency and low volatility, while incorrect ones demonstrate erratic and delayed convergence. Leveraging this, we introduce an intrinsic reward to distinguish the correctness of responses, along with a curiosity reward that promotes diverse exploration. Extensive experiments show that CoVo achieves performance on par with or even surpasses rule-based RL methods and demonstrates stability compared to other self-rewarding methods. Our method establishes a new paradigm for autonomous reasoning development in LLMs by grounding optimization on the inherent patterns of reasoning trajectories, which eliminates the need for external supervision.

## Acknowledgement

This work is supported in part by the Hangzhou Joint Funds of the Zhejiang Provincial Natural Science Foundation of China under Grant No. LHZSD24F020001, in part by the Zhejiang Province High-Level Talents Special Support Program "Leading Talent of Technological Innovation of Ten-Thousands Talents Program" under Grant No. 2022R52046, in part by the Fundamental Research Funds for the Central Universities under Grant No. 2021FZZX001-23. This work is supported by the Alibaba Research Intern Program. This research is supported by the RIE2025 Industry Alignment Fund – Industry Collaboration Projects (IAF-ICP) (Award I2301E0026), administered by A*STAR, as well as supported by Alibaba Group and NTU Singapore through Alibaba-NTU Global e-Sustainability CorpLab (ANGEL).

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

# Appendix

## Table of Contents

# A  More Experimental Results and Discussion

## A.1  Environmental Setups

**Evaluation Metric.** We evaluate both the *Performance* and *Reasoning Diversity* of CoVo in our experiments. For performance evaluation, We employ Pass@1 accuracy under a sampling temperature of 0 across all benchmark datasets as our metric, with correctness determined through comparison between generated outputs and ground truth. We also evaluate the model capacity ceiling using Pass@$k$ score on AIME-24 under a sampling temperature of 1. Our evaluation prompts on initial model across different reasoning domains are listed in Table 4. For diversity evaluation, we select Average Cosine Similarity (ACS) [36], Entropy [41] and Self-BLEU [75] as metrics. ACS measures the semantic similarity among different reasoning paths; Entropy quantifies the distributional variability over reasoning tokens; Self-BLEU assesses lexical overlap among samples. Together, the three metrics complement each other and can reflect the reasoning diversity beyond performance.

Table 4: Evaluation prompts on initial model across different reasoning domains.

| Domains | CoT Prompts |
|---|---|
| Mathematics | Question: {}\nPlease reason step by step, and put your final answer within \boxed{}. |
| Commonsense & Science | Question: {}\nAnswer the multiple choice question. The last line of your response should be of the following format: 'Answer: $LETTER' (without quotes) where LETTER is one of ABCD. Think step by step before answering. |

**Implementation Details.** All experiments run on 8×A100-80GB GPUs. We use OpenRLHF [16] to implement our algorithms for RL training. Here we provide detailed hyper-parameters in Table 5.

Table 5: Hyperparameters used in the training of CoVo.

| Settings | Hyperparameters |
|---|---|
| Sampling | top_k = -1 |
| | top_p = 1.0 |
| | temperature = 1.0 |
| | max_new_tokens = 4,096 |
| | n_samples_per_prompt = 16 |
| RL algorithm | Reinforce++ |
| Training | rollout_batch_size = 16 |
| | micro_rollout_batch_size = 16 |
| | train_batch_size = 32 |
| | micro_train_batch_size = 8 |
| | learning_rate = 5e-7 |
| | init_kl_coef = 1e-4 |
| | epochs = 1 |
| Optimization | adam_offload, packing_samples, |
| | flash_attn, colocate_all_models, |
| | vllm_enable_sleep, deepspeed_enable_sleep |

## A.2  Additional Results

**Larger Model Sizes.** To demonstrate the effectiveness of CoVo when model parameter scaling, we also conduct experiments on Qwen2.5-7B-Instruct and Qwen2.5-14B-Instruct [57]. The results in Table 6 and Fig. 6 right show that even with an increase in parameters, our method still achieves performance on par with or even surpassing the supervised RL baselines in most cases. Given the growing demand for large-scale RL in industry, this scalability demonstrates the effectiveness of our approach and its potential for broad real-world deployment.

**Reasoning Diversity.** In addition to performance improvements, we investigate whether our method promotes diverse reasoning strategies rather than collapsing to stereotypical solutions. We quanti-

Table 6: Results of all methods in diverse reasoning domains. *Italics* denotes that this method is training-free, "*" denotes that this method requires sampling multiple outputs for inferences, "†" denotes that this method utilizes self-rewarding without external labels, underline represents the best performance among all baselines, **bold** represents the best performance among all methods.

| **Model III: Qwen2.5-7B-Instruct (Accuracy ↑)** | | | | | | | |
|---|---|---|---|---|---|---|---|
| **Method** | MATH-500 | GSM8K | AMC-23 | Olympiad | MMLU-Pro | GPQA | CommonsenseQA |
| 1. *Zeroshot-CoT* | 74.4 | 91.1 | 55.0 | 39.4 | 55.6 | 36.2 | 81.3 |
| 2. *Fewshot-CoT* | 75.6 | 91.6 | 50.0 | 40.9 | 56.4 | 36.4 | 82.1 |
| 3. *CoT+SC@4 ** | 76.4 | 92.1 | 52.5 | 39.7 | 56.9 | 36.8 | 82.6 |
| 4. GRPO | 76.6 | 92.3 | 57.5 | **41.3** | 57.1 | 36.6 | **82.9** |
| 5. RLOO | 76.2 | 91.8 | 55.0 | 40.9 | **57.4** | 36.4 | 82.4 |
| 6. IRPO | 74.2 | 91.4 | 47.5 | 39.6 | 56.6 | 34.8 | 81.2 |
| 7. Reinforce | 76.4 | 92.3 | 57.5 | **41.3** | 56.9 | 36.8 | 82.7 |
| 8. Reinforce++ | 76.6 | 92.6 | **60.0** | 40.9 | 57.2 | 36.6 | 82.5 |
| 9. EMPO † | 76.4 | 92.5 | 55.0 | 41.2 | **57.4** | 37.1 | 82.8 |
| 10. TTRL † | 76.6 | 92.8 | 57.5 | 40.9 | 56.7 | 36.6 | 81.9 |
| **CoVo (Ours) †** | **76.8** | 92.5 | **60.0** | 41.2 | 57.0 | 36.8 | 82.8 |

Table 7: Diversity of reasoning path, **bold** represents the best performance of RL training methods.

| **Model: Qwen2.5-3B-Instruct (ACS ↓ / Entropy ↑ / Self-BLEU ↓)** | | | | | | | |
|---|---|---|---|---|---|---|---|
| **Method** | MATH | GSM8K | AMC | Olympiad Bench | MMLU-Pro | GPQA | CommonsenseQA |
| 1. CoT | 0.892 / 0.150 / 0.688 | 0.921 / 0.161 / 0.756 | 0.866 / 0.196 / 0.748 | 0.867 / 0.229 / 0.599 | 0.844 / 0.313 / 0.500 | 0.798 / 0.492 / 0.432 | 0.852 / 0.739 / 0.419 |
| 2. GRPO | 0.938 / 0.033 / 0.920 | 0.946 / 0.034 / 0.903 | 0.914 / 0.047 / 0.919 | 0.912 / 0.057 / 0.757 | 0.891 / 0.105 / 0.692 | 0.845 / 0.208 / 0.644 | 0.801 / 0.384 / 0.559 |
| 3. TTRL | 0.914 / 0.065 / 0.927 | 0.922 / 0.071 / 0.894 | 0.888 / 0.089 / 0.869 | **0.887** / 0.104 / **0.687** | 0.873 / 0.173 / 0.695 | 0.831 / 0.302 / 0.593 | 0.849 / 0.511 / **0.456** |
| **Ours** | **0.911 / 0.082 / 0.789** | **0.915 / 0.096 / 0.806** | **0.864 / 0.102 / 0.816** | 0.892 / **0.122** / 0.699 | **0.843 / 0.200 / 0.562** | **0.819 / 0.337 / 0.579** | **0.760 / 0.582** / 0.501 |

tatively analyze the diversity of reasoning paths across different domains trained after CoVo using three key metrics mentioned in the last section. The results in Table 7 show that the model diversity exceeds that of rule-based RL methods and self-rewarding baselines, indicating broader solution space exploration. More intuitively, we encode the model responses using Sentence Transformers [38] and apply UMAP [30] for 2D reduction and visualization. As depicted in Fig. 6 left, the reasoning paths of our method exhibit a more dispersed distribution, in contrast to the concentrated clusters observed in GRPO.

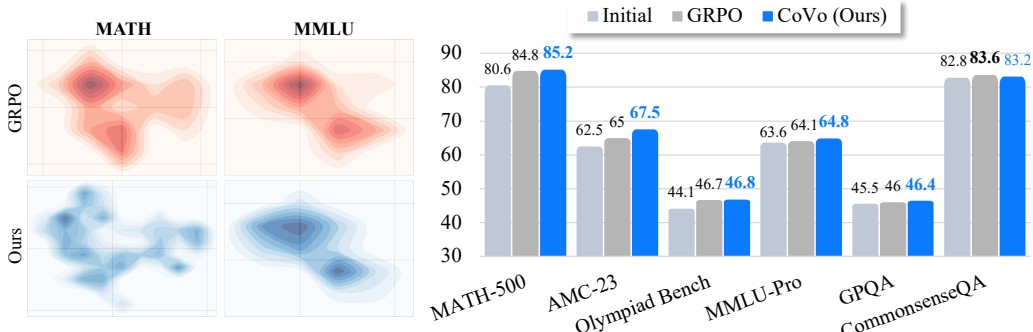

Figure 6: (Left) Visualization of reasoning diversity on different methods. (Right) Performance of Qwen2.5-14B-Instruct on different methods.

**Computational Resources.** In Section 5.3, we have qualitatively analyzed the computational complexity of CoVo. The computational overhead of our self-rewarding mechanism primarily stems from the calculation of distance matrices. Now we provide quantitative analysis on the reward computational time we need within a training step in Table 8. Our reward calculation only involves the prefilling stage, which is significantly faster than the decoding process. Since the distance matrix is computed independently and can take advantage of parallelism, the reward calculation indeed accounts for a small proportion of the time required in a

Table 8: Reward computational resources comparisons.

| **Method** | **Time per step** | **Proportion per step** |
|---|---|---|
| Ground Truth Reward | 2.06 s | 1.3% |
| Majority Voting Reward | 4.38 s | 2.8% |
| CoVo (Ours) | 49.22 s | 24.6% |

training step (nearly 24.6%). In the future, it will be possible to asynchronously compute the distance matrix during sampling to further reduce the time required for distance matrix computation.

## A.3 Discussion

### Q1: Why our methods have the potential to outperform rule-based RL?

First, rule-based RL methods do not assess the quality of the reasoning process. This poses a risk that incorrect reasoning paths may coincidentally lead to correct answers and thus receive undeserved high rewards. Secondly, we evaluate the correctness using intermediate reasoning states. In contrast, we place greater emphasis on the quality of the reasoning process rather than the final answer, which leads to more consistent improvements in reasoning ability.

### Q2: How does the method handle universally incorrect samples during RL?

Our reward mechanism may assign zero advantage to completely wrong answers due to the low consistency of intermediate reasoning states, which avoids misleading updates. This contrasts with majority voting, which amplifies dominant errors.

### Q3: Would CoVo have an implicit bias toward longer reasoning chains?

The core rewards in CoVo are defined based on the consistency between intermediate reasoning states and the final answer, not the length of the reasoning chains. Consistency measures how often intermediate states stay "close" to the final answer. A longer chain with redundant steps that still align with the final answer might maintain high consistency, but this is not rewarded because it is longer — it is rewarded because it remains coherent. Volatility measures how much intermediate states deviate from the final answer. A longer chain with erratic deviations (*e.g.*, the model contradicts itself before settling on an answer) would have high volatility and be penalized, regardless of length. Thus, our reward signal incentivizes stable, answer-aligned reasoning but is agnostic to the response length. The evolution of response length in Table 9 also supports this.

Table 9: The evolution of response length during the training process of Qwen2.5-3B-Instruct.

| Training Step | 0 | 50 | 100 | 150 | 200 | 250 | 300 | 350 | 400 | 450 |
|---|---|---|---|---|---|---|---|---|---|---|
| **Response Length (Token)** | 733 | 805 | 766 | 786 | 815 | 868 | 781 | 871 | 799 | 824 |

## A.4 Limitations and Future Work

The performance of RL-based method is inherently tied to the competence of the base policy. When the initial model exhibits fundamental limitations in task understanding or reasoning capacity, suboptimal outcomes are universally observed across RL methodologies, not uniquely in CoVo. This underscores a broader consensus in the field: the foundational capabilities of the base model serve as a critical prerequisite for effective RL fine-tuning. Advances in pre-training techniques have consistently demonstrated that stronger base models provide more robust starting points for subsequent RL optimization, a principle CoVo explicitly embraces. To validate the robustness of our framework against variations in initial policy quality, we conducted extensive experiments across model scales ranging from 3B to 14B parameters. Results show that CoVo yields consistent performance gains across all tested base models, including those with lower initial competence. This empirical evidence indicates that while CoVo, like all RL systems, benefits from stronger base policies, it maintains effectiveness and generalizability even when starting from less optimal initial states. Despite RL techniques, including CoVo, may not instill entirely new reasoning capabilities beyond what pre-training provides, their role in significantly enhancing the sampling efficiency and eliciting those pre-trained abilities is of paramount practical importance. We believe that self-rewarding RL methods like CoVo could prove valuable for downstream applications where human expert annotation of gold-standard answers is cost-prohibitive and unsustainable [68].

While our proposed method shows promising improvements in reasoning performance and training stability in LLMs, its generalizability to visual language models (VLMs) needs further exploration. Thus, our future research will focus on extending the self-rewarding mechanism to VLMs, where reasoning involves both linguistic and perceptual understanding. Additionally, we plan to investigate its effectiveness in semi-supervised scenarios with limited labels, which presents a more realistic and challenging scenario for practical applications.

# B Proof and Analysis

## B.1 Technical Setting and Notations

**Large Language Models.** For a prompt $x \in \mathcal{X}$, LLMs generate output distributions in an autoregressive manner, which can be formulated as:

$$\pi_\theta(y|x) = \prod_{i=1}^{|y|} \pi_\theta(y_i|x, y_{<i}) = \prod_{i=1}^{|y|} \mathrm{softmax}(f_\theta(x, y_{<i})), \tag{13}$$

where $f_\theta$ is a neural network that maps the sequence of tokens to the logits of the next token, and $y_{<l}$ denotes the prefix of the output sequence up to position $l - 1$. This parameterization corresponds to large-scale language models (*e.g.*, Transformers) used in practice. However, the non-concavity of the RL objective poses significant challenges for theoretical analysis [37, 1]. To simplify, existing studies [37, 31, 1] often consider a tabular policy that assigns an independent, trainable logit to each possible output $y$. The tabular policy can be regarded as a special case of the autoregressive policy, where the output distribution is directly parameterized, thus facilitating gradient analysis.

**Reinforcement Learning Object.** Let $\mathcal{V}$ be a finite vocabulary of tokens. We use $\pi_\theta$ to denote a language model, parameterized by $\theta$, that receives a prompt $x \in \mathcal{X}$ and produces a distribution over outputs $y \in \mathcal{Y}$, where $\mathcal{X}, \mathcal{Y} \subseteq \mathcal{V}^*$ are the sets of possible input prompts and outputs, respectively. Following convention in the RL literature, we also refer to $\pi_\theta$ as a policy. For a prompt $x \in \mathcal{X}$ and reward function $r : \mathcal{X} \times \mathcal{Y} \in \{0, 1\}$, we denote the RL objective for a single prompt $x \in \mathcal{X}$ by:

$$\mathcal{J}_{\mathrm{RL}}(\theta; x) = \mathbb{E}_{y \sim \pi_\theta(\cdot|x)}[r(x, y)]. \tag{14}$$

**Optimization.** We follow the settings in Razin et al. [37] and also adopt the gradient flow approximation of the policy gradient method, assuming an relatively small learning rate. The resulting dynamical system is described by:

$$\frac{d}{dt}\theta(t) = \nabla \mathcal{J}_{\mathrm{RL}}(\theta(t)), \quad t \geq 0, \tag{15}$$

where $\theta(t)$ denotes the parameters of the policy at timestep $t$ during training and $\theta(0)$ is $\theta_{\mathrm{ref}}$. The gradient flow formulation facilitates the analysis of the optimization dynamics by eliminating the stochastic noise inherent in gradient-based updates.

**Notations.** We use $\mathbf{e}_i$ to denote the $i$-th standard basis vector, $\|\cdot\|_2$ denotes the Euclidean ($\ell_2$) norm. $r^*$ represents real rewards derived from label, $r_{\mathrm{covo}}$ represents the reward function of CoVo.

Overall, our theoretical settings involve two main simplifications. First, we assume the KL coefficient is relatively small and will not be dominant. Second, the analysis is based on tabular policies.

## B.2 Useful Lemmas

**Lemma 1.** *Given a tabular policy $\pi_\theta$ and a prompt $x \in \mathcal{X}$, the following holds:*

$$\nabla \mathcal{J}_{RL}(\theta; x) = \pi_\theta(\cdot|x) \cdot r(x, \cdot) - \pi_\theta(\cdot|x) \cdot \mathbb{E}_{y \sim \pi_{\theta(0)}(\cdot|x)}[r(x, y)]. \tag{16}$$

*Proof.* We first compute the gradient of $\mathcal{J}_{\mathrm{RL}}(\theta; x)$ with respect to the parameter vector $\theta_{:,x}$, which corresponds to the column of parameters for prompt $x$:

$$\nabla \mathcal{J}_{\mathrm{RL}}(\theta; x) = \nabla \mathbb{E}_{y \sim \pi_\theta(\cdot|x)}[r(x, y)] = \mathbb{E}_{y \sim \pi_\theta(\cdot|x)}[r(x, y) \nabla \ln \pi_\theta(y|x)]. \tag{17}$$

Thus, we can write

$$\nabla \mathcal{J}_{\mathrm{RL}}(\theta; x) = \sum_{y \in \mathcal{Y}} \pi_\theta(y|x) \cdot r(x, y) \cdot \nabla \ln \pi_\theta(y|x). \tag{18}$$

According to the properties of tabular policy [44], we have $\nabla \ln \pi_\theta(y|x) = \mathbf{e}_y - \pi_\theta(\cdot|x)$. So $\nabla \mathcal{J}_{\mathrm{RL}}(\theta; x)$ can be expressed as:

$$\begin{aligned} \nabla \mathcal{J}_{\mathrm{RL}}(\theta; x) &= \sum_{y \in \mathcal{Y}} \pi_\theta(y|x) \cdot r(x, y) \cdot (\mathbf{e}_y - \pi_\theta(\cdot|x)) \\ &= \sum_{y \in \mathcal{Y}} \pi_\theta(y|x) \cdot r(x, y) \cdot \mathbf{e}_y - \sum_{y \in \mathcal{Y}} \pi_\theta(y|x) \cdot r(x, y) \cdot \pi_\theta(\cdot|x) \\ &= \pi_\theta(\cdot|x) \cdot r(x, \cdot) - \pi_\theta(\cdot|x) \cdot \mathbb{E}_{y \sim \pi_{\theta(0)}(\cdot|x)}[r(x, y)], \end{aligned} \tag{19}$$

which concludes the proof. $\qquad\square$

**Lemma 2.** *Assuming gradient flow is applied to maximize the RL objective over a set of prompts $\mathcal{X}$. Then, for any prompt $x \in \mathcal{X}$, outputs $y \subseteq \mathcal{Y}$, and time $T \geq 0$, such that the following holds:*

$$
\begin{aligned}
\frac{d}{dt}\pi_{\theta(t)}(y|x) = {}& \pi_{\theta(t)}(y|x)^2 \Big[ r(x,y) - \mathbb{E}_{y\sim\pi_{\theta(t)}(\cdot|x)}[r(x,y)] \Big] \\
& - \pi_{\theta(t)}(y|x) \sum_{y'\in\mathcal{Y}} \pi_{\theta(t)}(y'|x)^2 \Big[ r(x,y') - \mathbb{E}_{y\sim\pi_{\theta(t)}(\cdot|x)}[r(x,y)] \Big],
\end{aligned}
\tag{20}
$$

*and*

$$
\pi_{\theta(T)}(y|x) \leq \pi_{\theta(0)}(y|x) \cdot \exp\left(2T\right). \tag{21}
$$

*Proof.* Since $\frac{d}{dt}\pi_{\theta(t)}(y|x) = \pi_{\theta(t)}(y|x) \cdot \frac{d}{dt}\ln\pi_{\theta(t)}(y|x)$, applying the chain rule and Lemma 1 we get:

$$
\begin{aligned}
\frac{d}{dt}\pi_{\theta(t)}(y|x) ={}& \pi_{\theta(t)}(y|x) \cdot \Big\langle \nabla\ln\pi_{\theta(t)}(y|x), \frac{d}{dt}\theta(t) \Big\rangle \\
={}& \pi_{\theta(t)}(y|x) \cdot \Big\langle \mathbf{e}_y - \pi_{\theta(t)}(\cdot|x), \nabla\mathcal{J}_{\text{RL}}(\theta(t),x) \Big\rangle \\
={}& \pi_{\theta(t)}(y|x) \Big\langle \mathbf{e}_y - \pi_{\theta(t)}(\cdot|x), \pi_\theta(\cdot|x) \cdot \Big[ r(x,\cdot) - \mathbb{E}_{y\sim\pi_{\theta(0)}(\cdot|x)}[r(x,y)] \cdot \mathbf{1} \Big] \Big\rangle \\
={}& \underbrace{\pi_{\theta(t)}(y|x)^2 \Big[ r(x,y) - \mathbb{E}_{y\sim\pi_{\theta(t)}(\cdot|x)}[r(x,y)] \Big]}_{\text{(I)}} \\
& \underbrace{- \pi_{\theta(t)}(y|x) \sum_{y'\in\mathcal{Y}} \pi_{\theta(t)}(y'|x)^2 \Big[ r(x,y') - \mathbb{E}_{y\sim\pi_{\theta(t)}(\cdot|x)}[r(x,y)] \Big]}_{\text{(II)}}.
\end{aligned}
\tag{22}
$$

Since $|r(x,y) - \mathbb{E}_{y\sim\pi_{\theta(t)}(\cdot|x)}[r(x,y)]| \leq 1$, the value of (I) + (II) can be bounded as:

$$
\text{(I)} + \text{(II)} \leq \pi_{\theta(t)}(y|x)^2 + \pi_{\theta(t)}(y|x) \leq 2\pi_{\theta(t)}(y|x). \tag{23}
$$

Then we use Grönwall's inequality and derive:

$$
\frac{d}{dt}\pi_{\theta(t)}(y|x) \leq \pi_{\theta(t)}(y|x) \cdot 2 \implies \pi_{\theta(T)}(y|x) \leq \pi_{\theta(0)}(y|x) \cdot \exp\left(2T\right). \tag{24}
$$

Then we complete the proof. $\qquad\square$

### B.3 Monotonicity and Robustness of Intrinsic Reward

In Section 3.3, we propose intrinsic reward formulations $r_{\text{int}}^{L}$ and $r_{\text{int}}^{V}$ that employ two different feature combination strategies as reward signals. Specifically, the linear nature of $r_{\text{int}}^{L}$ effectively captures the monotonic characteristics of both the consistency ($Con(\tau)$) and volatility ($Vol(\tau)$) in a trajectory $\tau$. In contrast, the monotonicity of these two features in $r_{\text{int}}^{V}$ is not as pronounced. Now we formally prove that *$r_{\text{int}}^{V}$ and $r_{\text{int}}^{L}$ maintain equivalent monotonic dependencies on $Con(\tau)$ and $Vol(\tau)$, while $r_{\text{int}}^{V}$ exhibits greater robustness to outliers.* Inspired by the incremental method employed in Wang et al. [48] to establish monotonicity and robustness, we also adopt this idea for the proof.

We assume $n$ reasoning paths in a group, each represented as a pair of consistency and volatility $(C_i, V_i)$, where $1 \leq i \leq n$. The intrinsic reward for a group is denoted as $r_{\text{int}}(\mathbf{C}, \mathbf{V})$ with $\mathbf{C} = [C_i]_{i=1}^{n}$ and $\mathbf{V} = [V_i]_{i=1}^{n}$. Applying increments $\Delta c$ to $C_i$ and $\Delta v$ to $V_i$, the resulting change in the intrinsic reward is:

$$
\Delta r_{\text{int}}(C_i) = r_{\text{int}}(\mathbf{C} + \Delta c \cdot \mathbf{e}_i, \mathbf{V}) - r_{\text{int}}(\mathbf{C}, \mathbf{V}), \tag{25}
$$

$$
\Delta r_{\text{int}}(V_i) = r_{\text{int}}(\mathbf{C}, \mathbf{V} + \Delta v \cdot \mathbf{e}_i) - r_{\text{int}}(\mathbf{C}, \mathbf{V}). \tag{26}
$$

where $\mathbf{e}_i \in \mathbb{R}^n$ denotes the $i$-th standard basis vector with $i$-th component being 1 and the others 0.

### B.3.1 Monotonicity Analysis

For linear intrinsic reward, its group reward can be denoted as:

$$r_{\text{int}}^{\boldsymbol{L}}(\mathbf{C}, \mathbf{V}) = \sum_{j=1}^{n} \frac{C_i}{n} - \sum_{j=1}^{n} \frac{V_i}{n}, \tag{27}$$

Thus, the reward variations with feature increments are:

$$\Delta r_{\text{int}}^{\boldsymbol{L}}(C_i) = \frac{\Delta c}{n}, \; \Delta r_{\text{int}}^{\boldsymbol{L}}(V_i) = -\frac{\Delta v}{n}, \tag{28}$$

which ensures $\Delta r_{\text{int}}^{\boldsymbol{L}}(C_i) > 0$ and $\Delta r_{\text{int}}^{\boldsymbol{L}}(V_i) < 0$. That is, higher consistency and lower volatility indicate a higher $r_{\text{int}}^{\boldsymbol{L}}$ reward.

For vector-space intrinsic reward, its group reward can be denoted as:

$$r_{\text{int}}^{\boldsymbol{V}}(\mathbf{C}, \mathbf{V}) = \sqrt{\left(\frac{\sum_{j=1}^{n} C_j \cos V_j}{n}\right)^2 + \left(\frac{\sum_{j=1}^{n} C_j \sin V_j}{n}\right)^2}$$

$$= \frac{1}{n}\sqrt{\sum_j C_j^2 + \sum_{k,t,k \neq t} 2C_k C_t \cos V_k \cos V_t + \sum_{k,t,k \neq t} 2C_k C_t \sin V_k \sin V_t} \tag{29}$$

$$= \frac{1}{n}\sqrt{\sum_j C_j^2 + \sum_{k,t,k \neq t} 2C_k C_t \cos(V_k - V_t)}.$$

We denote $A = \sum_{k,t,k \neq t \neq i} 2C_k C_t \cos(V_k - V_t)$ and $B = \sum_{j,j \neq i} 2(C_i + \Delta c)C_j \cos(V_i - V_j)$, then the reward variations with feature increments are:

$$\Delta r_{\text{int}}^{\boldsymbol{V}}(C_i) = r_{\text{int}}^{\boldsymbol{V}}(\mathbf{C} + \Delta c \cdot \mathbf{e}_i, \mathbf{V}) - r_{\text{int}}^{\boldsymbol{V}}(\mathbf{C}, \mathbf{V})$$

$$= \frac{1}{n}\sqrt{\sum_{j=1,j \neq i} C_j^2 + (C_i + \Delta c)^2 + A + B} \; -$$

$$\frac{1}{n}\sqrt{\sum_j C_j^2 + \sum_{k,t,k \neq t} 2C_k C_t \cos(V_k - V_t)}$$

$$= \frac{-C_i^2 + (C_i + \Delta c)^2 + A - \sum_{k,t,k \neq t} 2C_k C_t \cos(V_k - V_t) + B)}{n^2 r_{\text{int}}^{\boldsymbol{V}}(\mathbf{C} + \Delta c \cdot \mathbf{e}_i, \mathbf{V}) + n^2 r_{\text{int}}^{\boldsymbol{V}}(\mathbf{C}, \mathbf{V})} \tag{30}$$

$$= \frac{-C_i^2 + (C_i + \Delta c)^2 + \sum_{j,j \neq i} 2C_j \Delta c \cos(V_i - V_j)}{n^2 r_{\text{int}}^{\boldsymbol{V}}(\mathbf{C} + \Delta c \cdot \mathbf{e}_i, \mathbf{V}) + n^2 r_{\text{int}}^{\boldsymbol{V}}(\mathbf{C}, \mathbf{V})}$$

$$= \frac{\Delta c\left(\Delta c + 2C_i + 2\sum_{j,j \neq i} C_j \cos(V_i - V_j)\right)}{n^2 r_{\text{int}}^{\boldsymbol{V}}(\mathbf{C} + \Delta c \cdot \mathbf{e}_i, \mathbf{V}) + n^2 r_{\text{int}}^{\boldsymbol{V}}(\mathbf{C}, \mathbf{V})},$$

and

$$\Delta r_{\text{int}}^{\boldsymbol{V}}(V_i) = r_{\text{int}}^{\boldsymbol{V}}(\mathbf{C}, \mathbf{V} + \Delta v \cdot \mathbf{e}_i) - r_{\text{int}}^{\boldsymbol{V}}(\mathbf{C}, \mathbf{V})$$

$$= \frac{1}{n}\sqrt{\sum_j L_j^2 + A + \sum_{j,j \neq i} 2C_i C_j \cos(V_i + \Delta v - V_j)} -$$

$$\frac{1}{n}\sqrt{\sum_j C_j^2 + \sum_{k,t,k \neq t} 2C_k C_t \cos(V_k - V_t)}$$

$$= \frac{A - \sum_{k,t,k \neq t} 2C_k C_t \cos(V_k - V_t) + \sum_{j,j \neq i} 2C_i C_j \cos(V_i + \Delta v - V_j)}{n^2 r_{\text{int}}^{\boldsymbol{V}}(\mathbf{C}, \mathbf{V} + \Delta v \cdot \mathbf{e}_i) + n^2 r_{\text{int}}^{\boldsymbol{V}}(\mathbf{C}, \mathbf{V})} \tag{31}$$

$$= \frac{\sum_{i,j,i \neq j} 2C_i C_j [\cos(V_i + \Delta v - V_j) - \cos(V_i - V_j)]}{n^2 r_{\text{int}}^{\boldsymbol{V}}(\mathbf{C}, \mathbf{V} + \Delta v \cdot \mathbf{e}_i) + n^2 r_{\text{int}}^{\boldsymbol{V}}(\mathbf{C}, \mathbf{V})}$$

$$= \frac{-4\sum_{i,j,i \neq j} C_i C_j \sin(V_i - V_j + \frac{\Delta v}{2})\sin(\frac{\Delta v}{2})}{n^2 r_{\text{int}}^{\boldsymbol{V}}(\mathbf{C}, \mathbf{V} + \Delta v \cdot \mathbf{e}_i) + n^2 r_{\text{int}}^{\boldsymbol{V}}(\mathbf{C}, \mathbf{V})}.$$

In practice, $V_i \in [0, 1]$ guarantees that $cos(V_i - V_j) > 0$, which always leads to $\Delta r_{\text{int}}^{V}(C_i) > 0$. When $\Delta v$ causes $V_i$ to deviate from the current group, it tends to be sizable. Therefore $(V_i - V_j + \frac{\Delta v}{2}) > 0$, which leads to $\Delta r_{\text{int}}^{V}(V_i) < 0$. Thus in our scenario, the monotonicity of $r_{\text{int}}^{V}$ on both magnitude and angle features is consistent with $r_{\text{int}}^{L}$.

### B.3.2 Robustness Analysis

In Section 3.3, we point out that $r_{\text{int}}^{L}$ may be more sensitive to outliers. Empirically, we validate this claim in the ablation study in Section 4.3. Here we provide the theoretical proof.

We already know that correct trajectories tend to exhibit higher consistency in intermediate reasoning states. Consequently, for a set of incorrect trajectories, the presence of abnormally large consistency values in some samples can lead to spurious reward assignments. Therefore, a reward signal capable of better controlling its increment under these circumstances would mitigate the risk of incorrect reward assignments. Therefore, we compare $\Delta r_{\text{int}}^{L}(C_i)$ and $\Delta r_{\text{int}}^{V}(C_i)$ under the same increments $\Delta c$, the smaller one indicates stronger robustness. We denote

$$r_{\text{int}}^{V}(\mathbf{C}, \mathbf{V}) = \frac{1}{n}\sqrt{\sum_{j} C_j^2 + \sum_{k,t,k \neq t} 2C_k C_t \cos(V_k - V_t)} = \frac{1}{n}\sqrt{S}, \tag{32}$$

where

$$\begin{aligned}
S &= \sum_{j} C_j^2 + \sum_{k,t,k \neq t} 2C_k C_t \cos(V_k - V_t) \\
&= \left( C_i + \sum_{j,j \neq i} C_j \cos(V_i - V_j) \right)^2 \\
&\quad + \sum_{j,j \neq i} C_j^2 + \sum_{k,t,k \neq t \neq i} 2C_k C_t \cos(V_k - V_t) - \left( \sum_{j,j \neq i} C_j \cos(V_i - V_j) \right)^2.
\end{aligned} \tag{33}$$

Note that

$$\begin{aligned}
&\sum_{j,j \neq i} C_j^2 + \sum_{k,t,k \neq t \neq i} 2C_k C_t \cos(V_k - V_t) - \left( \sum_{j,j \neq i} C_j \cos(V_i - V_j) \right)^2 \\
&= \sum_{j,j \neq i} C_j^2 + \sum_{k,t,k \neq t \neq i} 2C_k C_t \cos(V_k - V_t) \\
&\quad - \left( \sum_{j=1,j \neq i} C_j^2 \cos^2(V_i - V_j) + \sum_{k,t,k \neq t \neq i} 2C_k C_t \cos(V_i - V_k) \cos(V_i - V_t) \right) \\
&= \sum_{j=1,j \neq i} C_j^2 \left[ 1 - \cos^2(V_i - V_j) \right] \\
&\quad + \sum_{k,t,k \neq t \neq i} 2C_k C_t \left[ \cos(V_k - V_t) - \cos(V_i - V_k) \cos(V_i - V_t) \right] \\
&= \sum_{j=1,j \neq i} C_j^2 \left[ 1 - \cos^2(V_i - V_j) \right] \\
&\quad + \sum_{k,t,k \neq t \neq i} 2C_k C_t \left[ \cos((V_i - V_t) - (V_i - V_k)) - \cos(V_i - V_k) \cos(V_i - V_t) \right] \\
&= \sum_{j=1,j \neq i} C_j^2 \sin^2(V_i - V_j) + \sum_{k,t,k \neq t \neq i} 2C_k C_t \sin(V_i - V_k) \sin(V_i - V_t) \\
&= \left( \sum_{j=1,j \neq i} C_j \sin(V_i - V_j) \right)^2.
\end{aligned} \tag{34}$$

Therfore, we substitute Eq. (34) into Eq. (32) and obtain the following lower bound:

$$r_{\text{int}}^{\boldsymbol{V}}(\mathbf{C}, \mathbf{V}) = \frac{1}{n} \sqrt{\left(C_i + \sum_{j,j\neq i} C_j \cos(V_i - V_j)\right)^2 + \left(\sum_{j,j\neq i} C_j \sin(V_i - V_j)\right)^2}$$

$$\geq \frac{1}{n}\left(C_i + \sum_{j,j\neq i} C_j \cos(V_i - V_j)\right). \tag{35}$$

Then, we use this lower bound to deflate $\Delta r_{\text{int}}^{\boldsymbol{V}}(C_i)$ in Eq. (30):

$$\Delta r_{\text{int}}^{\boldsymbol{L}}(C_i) = \frac{\Delta c \left(\Delta c + 2C_i + 2\sum_{j,j\neq i} C_j \cos(V_i - V_j)\right)}{n^2 \mathcal{R}\left(\mathbf{C} + \Delta c \cdot \mathbf{e}_i, \mathbf{V}\right) + n^2 \mathcal{R}\left(\mathbf{C}, \mathbf{V}\right)}$$

$$\leq \frac{\Delta c \left(\Delta c + 2C_i + 2\sum_{j,j\neq i} C_j \cos(V_i - V_j)\right)}{n^2 \cdot \frac{1}{n}\left(C_i + \Delta c + \sum_{j,j\neq i} C_j \cos(V_i - V_j)\right) + n^2 \cdot \frac{1}{n}\left(C_i + \sum_{j,j\neq i} C_j \cos(V_i - V_j)\right)}$$

$$= \frac{\Delta c}{n}. \tag{36}$$

Note that the right side of Eq. (36) is exactly right side of $\Delta r_{\text{int}}^{\boldsymbol{L}}(C_i)$ in Eq. (28), which suggests $\Delta r_{\text{int}}^{\boldsymbol{V}}(C_i) \leq \Delta r_{\text{int}}^{\boldsymbol{L}}(C_i)$. According to the aforementioned analysis, we can conclude that $r_{\text{int}}^{\boldsymbol{V}}$ is more robust than $r_{\text{int}}^{\boldsymbol{L}}$ when facing abnormally large outliers.

### B.4 Proof of Proposition 1

**Proposition 1** (Model Collapse). *Given an input prompt $x \in \mathcal{X}$, the policy model sample $n$ responses $\{y_1, y_2, \ldots, y_n\} \subseteq \pi_\theta(\cdot|x)$ and denote $C(y) = \sum_{i=1}^{n} \mathbb{I}\{y_i = y\}$. Suppose $y^* \in \mathcal{Y}$ that satisfies $y^* = \operatorname{argmax}_{y \in \mathcal{Y}} \pi_\theta(y|x)$ and a reward function $r : \mathcal{X} \times \mathcal{Y}$ defined by*

$$r(x, y) = \begin{cases} 1, & \text{if } y = \operatorname{argmax}_{y' \in \mathcal{Y}} C(y'), \\ 0, & \text{otherwise.} \end{cases} \tag{37}$$

*such that the following hold:*

- *When using gradient ascent to maximize the RLHF objective with respect to the reward function $r$, $\pi_\theta(y^* \mid x)$ converges to 1 as training proceeds.*

- *If $y^*$ is not the ground truth, reward hacking will happen as training proceeds.*

*Proof.* We denote the RL object as:

$$\mathcal{J}_{RL}(\theta; x) = \mathbb{E}_{y \sim \pi_\theta(\cdot|x)}\left[r(x, y)\right], \tag{38}$$

By the log-derivative we derive the gradient:

$$\nabla \mathcal{J}_{\text{RL}}(\theta; x) = \mathbb{E}_{y \sim \pi_\theta(\cdot|x)}\left[r(x, y)\nabla \ln \pi_\theta(y|x)\right]$$

$$= \sum_{y \in \mathcal{Y}} \pi_\theta(y|x) \cdot r(x, y) \cdot \nabla \ln \pi_\theta(y|x). \tag{39}$$

We denote the parameters $\theta$ updates from timestep $t$ to $t + 1$ as:

$$\theta_{t+1} = \theta_t + \eta \cdot \nabla \mathcal{J}_{\text{RL}}(\theta; x) = \theta_t + \eta \cdot \sum_{y \in \mathcal{Y}} \pi_\theta(y|x) \cdot r(x, y) \cdot \nabla \ln \pi_\theta(y|x), \tag{40}$$

where $\eta$ is the learning rate. According to Taylor's formula, we obtain the following equation:

$$\ln \pi_{\theta(t+1)}(y|x) \approx \ln \pi_{\theta(t)}(y|x) + \nabla \ln \pi_{\theta(t)}(y|x)^T \cdot (\theta(t+1) - \theta(t)). \tag{41}$$

By substituting Eq. (40) into Eq. (41) we can get the log-probability for certain output $y^*$:

$$\ln \pi_{\theta(t+1)}(y^*|x) \approx \ln \pi_{\theta(t)}(y^*|x) + \eta \nabla \ln \pi_{\theta(t)}(y^*|x)^T \cdot \sum_{y \in \mathcal{Y}} \pi_\theta(y|x) r(x,y) \nabla \ln \pi_\theta(y|x)$$

$$= \ln \pi_{\theta(t)}(y^*|x) + \eta \Big( \pi_\theta(y^*|x) r(x,y^*) \nabla \ln \pi_\theta(y^*|x) \cdot \nabla \ln \pi_\theta(y^*|x)^T$$

$$+ \sum_{y,y \neq y^*} \pi_\theta(y|x) r(x,y) \nabla \ln \pi_\theta(y|x) \cdot \nabla \ln \pi_\theta(y^*|x)^T \Big) \tag{42}$$

$$= \ln \pi_{\theta(t)}(y^*|x) + \eta \Big( \underbrace{\pi_\theta(y^*|x) r(x,y^*) \|\nabla \ln \pi_\theta(y^*|x)\|_2^2}_{\text{(I)}}$$

$$+ \underbrace{\sum_{y,y \neq y^*} \pi_\theta(y|x) r(x,y) \nabla \ln \pi_\theta(y|x) \cdot \nabla \ln \pi_\theta(y^*|x)^T}_{\text{(II)}} \Big)$$

In our setup, $y^*$ is defined as the response that has the highest count $C(y)$ among the $n$ sampled responses, which means $r(x,y^*) = 1$ and $r(x,y) = 0$ for $y \neq y^*$. Therefore, term (I) is non-negative and term (II) is 0, which indicates the value of (I) + (II) is larger than 0.

Since $\pi(y^* \mid x) \leq 1$, we have $\ln \pi_{\theta(t)}(y^* \mid x) \leq \ln 1 = 0$. Thus the sequence $\{\ln \pi_{\theta(t)}(y^* \mid x)\}_{t=0}^{\infty}$ is monotonically increasing and bounded above by 0. By the Monotone Convergence Theorem, $\{\ln \pi_{\theta(t)}(y^* \mid x)\}$ will converge to its upper bound 0:

$$\lim_{t \to \infty} \ln \pi_{\theta(t)}(y^* \mid x) = 0 \quad \implies \quad \lim_{t \to \infty} \pi_{\theta(t)}(y^* \mid x) = 1. \tag{43}$$

To conclude, if $y^*$ is not the "ground-truth" desired output but merely the sample-mode that maximizes $C(y)$, the above shows that the policy will collapse to always output $y^*$, regardless of whether it semantically or factually answers the prompt correctly. In other words, the model has "hacked" the reward by finding a high-frequency sample that maximizes rewards $r$, rather than by producing the true correct response. □

### B.5 Proof of Proposition 2

We aim to establish a variational lower bound for the conditional likelihood $\log \pi_\theta(y|x)$, by introducing a latent variable $s$ representing latent reasoning paths. The idea is to show that optimizing this lower bound corresponds to assigning rewards to consistent reasoning paths, enabling the model to learn not only answer generation but also the intermediate states leading to those answers.

**Proposition 2** (Variational Optimization for Reasoning). *Let $\pi_\theta(y|x)$ denote a language model that generates final answers $y$ conditioned on input prompt $x$, with $s$ representing latent intermediate reasoning states. Suppose the reasoning state is guided by a prior distribution $\pi_{\theta(0)}(s|x)$ and define the reward function $r(s,x,y) \in [0,1] \propto \log \pi_\theta(y|x \oplus s)$, the following optimizing bound holds:*

$$\log \pi_\theta(y|x) \geq \underbrace{\mathbb{E}_{s,y \sim \pi_\theta(\cdot|x)} [r(s,y,x)]}_{\text{Reward Assignment}} - \underbrace{D_{KL} [\pi_\theta(s|x) \| \pi_{\theta(0)}(s|x)]}_{\text{KL Regularization}}, \tag{44}$$

*where $\oplus$ denotes the concatenation of the input prompt $x$ with the reasoning state $s$.*

*Proof.* We first introduce the latent variable $s$ as latent reasoning states. We begin by expressing the marginal likelihood of the output $y$ conditioned on input $x$ by marginalizing over reasoning states $s$ with respect to a prior distribution $\pi_{\theta(0)}(s|x)$:

$$\log \pi_\theta(y|x) = \log \int \pi_\theta(y|x,s) \pi_{\theta(0)}(s|x) \, ds, \tag{45}$$

where $\pi_\theta(y|x,s)$ denotes the probability of generating output $y$ given input $x$ augmented with the intermediate reasoning states $s$, *i.e.*, $\pi_\theta(y|x,s) := \pi_\theta(y|x \oplus s)$.

After that, we introduce a variational distribution $\pi_\theta(s|x)$ over the latent states and rewrite the integral using importance sampling:

$$
\begin{aligned}
\log \pi_\theta(y|x) &= \log \int \frac{\pi_\theta(s|x)}{\pi_\theta(s|x)} \pi_\theta(y|x,s) \pi_{\theta(0)}(s|x)\, ds \\
&= \log \mathbb{E}_{s,y \sim \pi_\theta(\cdot|x)} \left[ \frac{\pi_\theta(y|x,s) \pi_{\theta(0)}(s|x)}{\pi_\theta(s|x)} \right].
\end{aligned}
\tag{46}
$$

Applying Jensen's inequality yields the evidence lower bound (ELBO):

$$
\log \pi_\theta(y|x) \geq \mathbb{E}_{s,y \sim \pi_\theta(\cdot|x)} \left[ \log \pi_\theta(y|x,s) + \log \pi_{\theta(0)}(s|x) - \log \pi_\theta(s|x) \right],
\tag{47}
$$

which simplifies to:

$$
\log \pi_\theta(y|x) \geq \mathbb{E}_{s,y \sim \pi_\theta(\cdot|x)} \left[ \log \pi_\theta(y|x \oplus s) \right] - D_{\mathrm{KL}} \left[ \pi_\theta(s|x) \| \pi_{\theta(0)}(s|x) \right].
\tag{48}
$$

Recall that the reward function is defined as: $r(s,x,y) \propto \log \pi_\theta(y|x \oplus s)$. Without affecting the optimization, $\log \pi_\theta(y|x \oplus s)$ can be directly regarded as part of the reward function (since the constant ratio will not affect the final learning goal), then the above bound yields:

$$
\log \pi_\theta(y|x) \geq \mathbb{E}_{s,y \sim \pi_\theta(\cdot|x)} \left[ r(s,x,y) \right] - D_{\mathrm{KL}} \left[ \pi_\theta(s|x) \| \pi_{\theta(0)}(s|x) \right],
\tag{49}
$$

which completes the derivation. This formulation shows that optimizing the output likelihood can be interpreted as assigning rewards to latent reasoning states while regularizing their distribution toward a prior. Hence, self-rewarding reinforcement learning corresponds to performing approximate variational inference over reasoning trajectories, thereby coupling answer generation and reasoning trajectory inference in a principled manner. □

### B.6 Proof of Proposition 3

To analyze the convergence of CoVo, we establish the following proposition. The core idea of this proof is adapted from Razin et al. [37], with simplifications tailored to the context of our work.

**Proposition 3** (Convergence). *Let $\mathcal{X}$ denote prompt sets, and $\gamma > 0$ represent the expected increase of real reward $r^*$. For a prompt $x \in \mathcal{X}$, let $y^\gamma$ denote the final answer in a reasoning trajectory with higher consistency and lower volatility. Define $T'$ as the initial time where $\mathbb{E}_{y \sim \pi_\theta(\cdot|x)}[r^*(x,y)] \geq \mathbb{E}_{y \sim \pi_{\theta(0)}(\cdot|x)}[r^*(x,y)] + \gamma$. We assume the following conditions hold:*

- *Our reward function ranks $y^\gamma$ highest among all candidates;*

- *Our reward function has a low probability of misclassifying incorrect trajectories;*

- *The KL regularization coefficient is sufficiently small.*

*Then the convergence time for achieving the expected increase in real reward satisfies the upper bound $T' \leq \mathcal{O}(\pi_{\theta(0)}(y^\gamma|x)^{-1})$.*

*Proof.* We first formalize these assumptions. Define $\mathcal{Y}^+ \subseteq \mathcal{Y}$ be a set of outputs such that each $y^\gamma \in \mathcal{Y}^+$ s.t. $r^*(x,y^\gamma) > \mathbb{E}_{y \sim \pi_{\theta(0)}(\cdot|x)}[r^*(x,y)] + \gamma$ and $r_{\mathrm{covo}}(x,y^\gamma) > \mathbb{E}_{y \sim \pi_{\theta(0)}(\cdot|x)}[r_{\mathrm{covo}}(x,y)]$, $\mathcal{Y}^- \subseteq \mathcal{Y}$ be a set of outputs s.t. $\mathcal{Y}^- = \{y' \in \mathcal{Y} \mid r_{\mathrm{covo}}(x,y') > \mathbb{E}_{y \sim \pi_{\theta(0)}(\cdot|x)}[r_{\mathrm{covo}}(x,y)]\} \backslash \mathcal{Y}^+$. Assume that the initial probability of $\mathcal{Y}^-$ is small:

$$
\pi_{\theta(0)}(\mathcal{Y}^-|x) \leq \frac{\sigma(1-\rho)\pi_{\theta(0)}(\mathcal{Y}^+|x)^2}{4|\mathcal{Y}^+|} \exp(-2T'),
\tag{50}
$$

where $\rho = \mathbb{E}_{y \sim \pi_{\theta(0)}(\cdot|x)}[r^*(x,y)] + \gamma \in [0,1]$ and $\sigma = (1-\rho)(\max_{y \in \mathcal{Y}^+} r_{\mathrm{covo}}(x,y) - \mathbb{E}_{y \sim \pi_{\theta(0)}(\cdot|x)}[r_{\mathrm{covo}}(x,y)]) = (1-\rho)(1 - \mathbb{E}_{y \sim \pi_{\theta(0)}(\cdot|x)}[r_{\mathrm{covo}}(x,y)])$.

Since

$$
\begin{aligned}
\mathbb{E}_{y \sim \pi_{\theta(t)}(\cdot|x)}[r^*(x,y)] &= \pi_{\theta(t)}(\mathcal{Y}^+|x) \cdot 1 - (1 - \pi_{\theta(t)}(\mathcal{Y}^+|x)) \cdot 0 \\
&\geq \rho = \mathbb{E}_{y \sim \pi_{\theta(0)}(\cdot|x)}[r^*(x,y)] + \gamma,
\end{aligned}
\tag{51}
$$

we can find that $\pi_{\theta(t)}(\mathcal{Y}^+|x) \geq \rho$ implies $\mathbb{E}_{y \sim \pi_{\theta(t)}(\cdot|x)}[r^*(x,y)] \geq \mathbb{E}_{y \sim \pi_{\theta(0)}(\cdot|x)}[r^*(x,y)] + \gamma$. Thus, it suffices to show that $\pi_{\theta(t)}(\mathcal{Y}^+|x)$ reaches $\rho$ within a finite time $T'$.

For $t \in [0, T')$ and $y \in \mathcal{Y}^+$, by Lemma 2 we know:

$$
\begin{aligned}
\frac{d}{dt}\pi_{\theta(t)}(y|x) = &\ \pi_{\theta(t)}(y|x)^2 \left[ r_{\text{covo}}(x,y) - \mathbb{E}_{y \sim \pi_{\theta(t)}(\cdot|x)}[r_{\text{covo}}(x,y)] \right] \\
&- \pi_{\theta(t)}(y|x) \sum_{y' \in \mathcal{Y}} \pi_{\theta(t)}(y'|x)^2 \left[ r_{\text{covo}}(x,y') - \mathbb{E}_{y \sim \pi_{\theta(t)}(\cdot|x)}[r_{\text{covo}}(x,y)] \right].
\end{aligned}
\tag{52}
$$

Summing all outputs in $\mathcal{Y}^+$ then derives the following:

$$
\begin{aligned}
\frac{d}{dt}\pi_{\theta(t)}(\mathcal{Y}^+|x) = &\ \sum_{y^\gamma \in \mathcal{Y}^+} \pi_{\theta(t)}(y|x)^2 \left[ r_{\text{covo}}(x,y^\gamma) - \mathbb{E}_{y \sim \pi_{\theta(t)}(\cdot|x)}[r_{\text{covo}}(x,y)] \right] \\
&- \pi_{\theta(t)}(\mathcal{Y}^+|x) \sum_{y' \in \mathcal{Y}} \pi_{\theta(t)}(y'|x)^2 \left[ r_{\text{covo}}(x,y') - \mathbb{E}_{y \sim \pi_{\theta(t)}(\cdot|x)}[r_{\text{covo}}(x,y)] \right] \\
= &\ (1 - \pi_{\theta(t)}(\mathcal{Y}^+|x)) \sum_{y \in \mathcal{Y}^+} \pi_{\theta(t)}(y|x)^2 \underbrace{\left[ r_{\text{covo}}(x,y) - \mathbb{E}_{y \sim \pi_{\theta(t)}(\cdot|x)}[r_{\text{covo}}(x,y)] \right]}_{(A)} \\
&- \pi_{\theta(t)}(\mathcal{Y}^+|x) \sum_{y' \in \mathcal{Y}\backslash\mathcal{Y}^+} \pi_{\theta(t)}(y'|x)^2 \underbrace{\left[ r_{\text{covo}}(x,y') - \mathbb{E}_{y \sim \pi_{\theta(t)}(\cdot|x)}[r_{\text{covo}}(x,y)] \right]}_{(B)}.
\end{aligned}
\tag{53}
$$

To derive the lower bound of $\frac{d}{dt}\pi_{\theta(t)}(y|x)$, we begin to analyze the terms $A$ and $B$ above.

First, the value of term $A$ is:

$$
\begin{aligned}
A &= r_{\text{covo}}(x,y) - \mathbb{E}_{y \sim \pi_{\theta(t)}(\cdot|x)}[r_{\text{covo}}(x,y)] \\
&= r_{\text{covo}}(x,y) \sum_{y' \in \mathcal{Y}} \pi_{\theta(t)}(y'|x) - \sum_{y' \in \mathcal{Y}} \pi_{\theta(t)}(y'|x)\, r_{\text{covo}}(x,y') \\
&= \sum_{y' \in \mathcal{Y}} \pi_{\theta(t)}(y'|x) \left( r_{\text{covo}}(x,y) - r_{\text{covo}}(x,y') \right) \\
&= \sum_{y' \in \mathcal{Y}\backslash\mathcal{Y}^+} \pi_{\theta(t)}(y'|x) \left( r_{\text{covo}}(x,y) - r_{\text{covo}}(x,y') \right).
\end{aligned}
\tag{54}
$$

Through the definitions of $\mathcal{Y}^+$ and $\mathcal{Y}^-$, for any $y' \in \mathcal{Y}\backslash(\mathcal{Y}^+ \cup \mathcal{Y}^-)$, we know that $r_{\text{covo}}(x,y') \leq \mathbb{E}_{y \sim \pi_{\theta(0)}(\cdot|x)}[r_{\text{covo}}(x,y)]$, which impiles $r_{\text{covo}}(x,y) - r_{\text{covo}}(x,y') \geq r_{\text{covo}}(x,y) - \mathbb{E}_{y \sim \pi_{\theta(0)}(\cdot|x)}[r_{\text{covo}}(x,y)]$. Thus we can derive the following inequality:

$$
\begin{aligned}
A &= \sum_{y' \in \mathcal{Y}\backslash\mathcal{Y}^+} \pi_{\theta(t)}(y'|x)(r_{\text{covo}}(x,y) - r_{\text{covo}}(x,y')) \\
&= \sum_{y' \in \mathcal{Y}^-} \pi_{\theta(t)}(y'|x)(r_{\text{covo}}(x,y) - r_{\text{covo}}(x,y')) \\
&\quad + \sum_{y' \in \mathcal{Y}\backslash(\mathcal{Y}^+\cup\mathcal{Y}^-)} \pi_{\theta(t)}(y'|x)(r_{\text{covo}}(x,y) - r_{\text{covo}}(x,y')) \\
&\geq \sum_{y' \in \mathcal{Y}^-} \pi_{\theta(t)}(y'|x)(r_{\text{covo}}(x,y) - r_{\text{covo}}(x,y')) \\
&\quad + \sum_{y' \in \mathcal{Y}\backslash(\mathcal{Y}^+\cup\mathcal{Y}^-)} \pi_{\theta(t)}(y'|x)(r_{\text{covo}}(x,y) - \mathbb{E}_{y \sim \pi_{\theta(0)}(\cdot|x)}[r_{\text{covo}}(x,y)]) \\
&= \sum_{y' \in \mathcal{Y}^-} \pi_{\theta(t)}(y'|x)(\mathbb{E}_{y \sim \pi_{\theta(0)}(\cdot|x)}[r_{\text{covo}}(x,y)] - r_{\text{covo}}(x,y')) \\
&\quad + (1 - \pi_{\theta(t)}(\mathcal{Y}^+|x))(r_{\text{covo}}(x,y) - \mathbb{E}_{y \sim \pi_{\theta(0)}(\cdot|x)}[r_{\text{covo}}(x,y)]).
\end{aligned}
\tag{55}
$$

Since we assume that $\pi_{\theta(t)}(\mathcal{Y}^+|x)$ reaches $\rho$ at timestep $T'$, $\pi_{\theta(t)}(\mathcal{Y}^+|x) < \rho$ for $t \in [0, T')$. We also have $|\mathbb{E}_{y \sim \pi_{\theta(0)}(\cdot|x)}[r_{\mathrm{covo}}(x,y)] - r_{\mathrm{covo}}(x,y')| \leq 1$ for $y \in \mathcal{Y}$, we obtain:

$$A \geq (1-\rho)(r_{\mathrm{covo}}(x,y) - \mathbb{E}_{y \sim \pi_{\theta(0)}(\cdot|x)}[r_{\mathrm{covo}}(x,y)]) - \pi_{\theta(t)}(\mathcal{Y}^-|x)$$
$$= \sigma - \pi_{\theta(t)}(\mathcal{Y}^-|x). \tag{56}$$

Second, the value of term $B$ is:

$$B = \pi_{\theta(t)}(y'|x)^2 \left[ r_{\mathrm{covo}}(x,y') - \mathbb{E}_{y \sim \pi_{\theta(t)}(\cdot|x)}[r_{\mathrm{covo}}(x,y)] \right]$$
$$= \pi_{\theta(t)}(y'|x)^2 \sum_{y \notin \mathcal{Y}^-} \pi_{\theta(t)}(y|x)(r_{\mathrm{covo}}(x,y') - r_{\mathrm{covo}}(x,y))$$
$$+ \pi_{\theta(t)}(y'|x)^2 \sum_{y \in \mathcal{Y}^-} \pi_{\theta(t)}(y|x)(r_{\mathrm{covo}}(x,y') - r_{\mathrm{covo}}(x,y)). \tag{57}$$

According to the definition of $\mathcal{Y}^-$, we have $r_{\mathrm{covo}}(x,y') - r_{\mathrm{covo}}(x,y) = 0$ for any $y \notin \mathcal{Y}^-$ and $r_{\mathrm{covo}}(x,y') - \mathbb{E}_{y \sim \pi_{\theta(t)}(\cdot|x)}[r_{\mathrm{covo}}(x,y)] \leq 1$ for $y \in \mathcal{Y}^-$, the above equation leads to:

$$B \leq \pi_{\theta(t)}(y'|x)^2 \leq \pi_{\theta(t)}(y'|x). \tag{58}$$

By deriving the lower bound and upper bound of terms $A$ and $B$ respectively, we obtain the lower bound of Eq. (53):

$$\frac{d}{dt}\pi_{\theta(t)}(\mathcal{Y}^+|x) \geq (1-\rho) \sum_{y \in \mathcal{Y}^+} \pi_{\theta(t)}(y|x)^2(\sigma - \pi_{\theta(t)}(\mathcal{Y}^-|x)) - \rho \cdot \pi_{\theta(t)}(\mathcal{Y}^-|x)$$
$$\geq \frac{(1-\rho)}{|\mathcal{Y}^+|}(\sigma - \pi_{\theta(t)}(\mathcal{Y}^-|x))\pi_{\theta(t)}(\mathcal{Y}^+|x)^2 - \pi_{\theta(t)}(\mathcal{Y}^-|x). \tag{59}$$

From Lemma 2, we know that

$$\pi_{\theta(t)}(\mathcal{Y}^-|x) \leq \pi_{\theta(0)}(\mathcal{Y}^-|x) \cdot \exp(2T') = \frac{\sigma(1-\rho)\pi_{\theta(0)}(\mathcal{Y}^+|x)^2}{4|\mathcal{Y}^+|} \leq \frac{\sigma}{4} \leq \frac{\sigma}{2}. \tag{60}$$

Substituting the above inequality into Eq. (59), we obtain:

$$\frac{d}{dt}\pi_{\theta(t)}(\mathcal{Y}^+|x) \geq \frac{(1-\rho)\sigma}{4|\mathcal{Y}^+|}\pi_{\theta(t)}(\mathcal{Y}^+|x)^2. \tag{61}$$

Denote $\pi(t) = \pi_{\theta(t)}(\mathcal{Y}^+ \mid x)$, we can rewrite Eq. (61) as:

$$\frac{d\pi(t)}{dt} \geq \frac{(1-\rho)\sigma}{4|\mathcal{Y}^+|}\pi(t)^2. \tag{62}$$

We divide both sides of the equation by $\pi(t)^2$ and get:

$$\frac{1}{\pi(t)^2}\frac{d\pi(t)}{dt} \geq \frac{(1-\rho)\sigma}{4|\mathcal{Y}^+|} \quad \Longrightarrow \quad \frac{d}{dt}\left(\frac{1}{\pi(t)}\right) \geq \frac{(1-\rho)\sigma}{4|\mathcal{Y}^+|}. \tag{63}$$

Next, we integrate both sides of the inequality. Assume that $\pi(t)$ evolves from its initial value $\pi(0)$ at time $t = 0$ to the value $\rho$ at some time $t = T'$. The integration becomes:

$$\int_0^{T'} \frac{d}{dt}[\pi(t)^{-1}] \, dt \geq \int_0^{T'} \frac{(1-\rho)\sigma}{4|\mathcal{Y}^+|} \, dt. \tag{64}$$

Substituting the results of both integrals into the inequality, we get:

$$\frac{1}{\pi(0)} - \frac{1}{\rho} \geq \frac{(1-\rho)\sigma}{4|\mathcal{Y}^+|}T'. \tag{65}$$

Finally, solving for $T'$ we obtain:

$$T' \leq \frac{4|\mathcal{Y}^+|}{(1-\rho)\sigma}\left(\frac{1}{\pi(0)} - \frac{1}{\rho}\right) = \frac{4|\mathcal{Y}^+|}{(1-\rho)\sigma}\left(\frac{1}{\pi_{\theta(0)}(\mathcal{Y}^+|x)} - \frac{1}{\rho}\right) = \mathcal{O}(\pi_0(y^\gamma|x)^{-1}). \tag{66}$$

This inequality indicates that when $t = T'$, $\pi_{\theta(t)}(\mathcal{Y}^+|x) \geq \rho$, which means the increase by $\gamma$ of the real reward. Then we complete the proof. $\qquad\square$

---

**Algorithm 1** Intrinsic Reward Computation

---

**Input:** $G$: The number of trajectories in a group with the same answer

$\quad\quad \boldsymbol{D_1}, \boldsymbol{D_2}, ..., \boldsymbol{D_G} \in \mathbb{R}^{T \times K}$: Distance Matrix of all trajectories

1: $r_{\text{int}}^L \leftarrow 0$
2: **for** $i \leftarrow 0$ **to** $G - 1$ **do**
3: $\quad$ Con $\leftarrow 0$
4: $\quad$ Vol $\leftarrow 0$
5: $\quad \boldsymbol{V_G} \leftarrow \boldsymbol{0}$
6: $\quad$ **for** $t \leftarrow 0$ **to** $T - 1$ **do**
7: $\quad\quad$ **if** $\boldsymbol{D}[t, 0] == \min \boldsymbol{D}[t, :]$ **then**
8: $\quad\quad\quad$ Con $\leftarrow$ Con $+ 1/T$
9: $\quad\quad$ **else**
10: $\quad\quad\quad$ Vol $\leftarrow t/T$
11: $\quad\quad$ **end if**
12: $\quad$ **end for**
13: $\quad r_{\text{int}}^L \leftarrow r_{\text{int}}^L + \text{Con} - \text{Vol}$
14: $\quad \boldsymbol{V_G} \leftarrow \boldsymbol{V_G} + [\text{Con} \cdot \cos(\text{Vol}), \text{Con} \cdot \sin(\text{Vol})]$
15: **end for**
16: $r_{\text{int}}^L \leftarrow r_{\text{int}}^L / G$
17: $r_{\text{int}}^V \leftarrow \|\boldsymbol{V_G}\|_2 / G$
**Output:** $r_{\text{int}}^L, r_{\text{int}}^V$

---

---

**Algorithm 2** Curiosity Reward Computation

---

**Input:** $T$: The number of intermediate reasoning states in a trajectory

$\quad\quad s_1, s_2, ..., s_T$: All intermediate reasoning states

1: $r_{\text{cur}} \leftarrow 0$
2: **for** $i \leftarrow 0$ **to** $T - 1$ **do**
3: $\quad$ Distance $\leftarrow 0$
4: $\quad \mathcal{P} = []$
5: $\quad$ **for** $t \leftarrow \text{len}(s_i)$ **to** $\text{len}(s_{i+1})$ **do**
6: $\quad\quad$ logprob $= \log \pi_\theta(s_{i+1}[t] | s_{i+1}[:t])$
7: $\quad\quad$ Distance $\leftarrow$ Distance $+$ logprob
8: $\quad\quad \mathcal{P}.\text{append(logprob)}$
9: $\quad$ **end for**
10: $\quad$ Distance $\leftarrow$ Distance$/(\text{len}(s_{i+1}) - \text{len}(s_i))$
11: $\quad$ KLP $= \ln[\text{KL}(\mathcal{P}, \mathcal{U}) + 1]$
12: $\quad r_{\text{cur}} \leftarrow r_{\text{cur}} + \text{Distance} - \text{KLP}$
13: **end for**
14: $r_{\text{cur}} \leftarrow r_{\text{cur}} / T$
**Output:** $r_{\text{cur}}$

---

## C  Algorithm: Reward Computation

To facilitate reproducibility, we provide the detailed algorithm for computing intrinsic reward and curiosity reward in Algorithm 1 and Algorithm 2, respectively.

## D  Datasets Details

We select six datasets in three different domains for evaluation in our experiments, and list details about our evaluation datasets in Table 10. We briefly describe the information of each dataset.

### D.1  Training

The training dataset of Open-Reasoner-Zero [17] comprises public data from various sources, including AIME (up to 2023), MATH [14], Numina-Math collection [24], Tulu3 MATH [23], OpenR1-Math-220k [2] and other opensource datasets. It also includes additional synthesized reasoning tasks

using programmatic approaches to augment the dataset. Note that most of our training dataset was released after Qwen2.5 series, making our training data a comparatively clean starting point for RL experiments to avoid potential data contamination [53].

## D.2 Evaluation

- Mathematical Domain: **MATH-500** [26] dataset contains 500 high school–level math problems, covering 7 major areas such as precalculus, algebra, and number theory. **GSM8K** [7] contains 1319 elementary school math problems, each requiring 2 to 8 steps to solve and mainly involving basic arithmetic operations. The **AMC–23**[3] dataset contains 40 high school–level math competition problems, which are challenging and diverse in form. **Olympiad Bench** [12] includes challenging math problems sourced from international Olympiads, Chinese Olympiads, and the Chinese College Entrance Examination.

- Commonsense Domain: **MMLU-Pro** [50] covers a wide range of fields, including natural sciences, social sciences, humanities, and some interdisciplinary content, which can effectively assess both the knowledge base and reasoning abilities of LLMs. **CommonsenseQA** [45] mainly includes a series of commonsense questions involving various aspects such as daily life, social culture, and natural phenomena, which aim to assess LLM's understanding of commonsense knowledge.

- Science Domain: **GPQA** [39] is a multiple-choice, Q&A dataset of very hard questions written and validated by experts in biology, physics, and chemistry, which measures LLM's knowledge and reasoning capabilities in scientific fields.

Table 10: The detailed information of seven evaluation datasets.

| Dataset | # Train | # Test | # Domain |
|---|---|---|---|
| MATH | 7500 | 5000 | Mathematics |
| GSM8K | 7473 | 1319 | Mathematics |
| Olympiad Bench | - | 675 | Mathematics |
| AMC | - | 40 | Mathematics |
| MMLU-Pro | - | 12032 | Commonsense |
| CommonsenseQA | 9741 | 1140 | Commonsense |
| GPQA | - | 448 | Science |

# E Case study

Here we provide a detailed reasoning case showing the reasoning trajectories change before and after training.

---

**Case study: Initial Qwen2.5-3B-Instruct**

\# Problem: The coordinates of a parallelogram are (5, 3), (6, 8), (7, 4) and $(x, y)$ and $x > 7$. What is the value of $x + y$?

\# Output: To determine the coordinates of the fourth vertex of the parallelogram, we need to utilize the property that the diagonals of a parallelogram bisect each other. This means that the midpoint of one diagonal is the same as the midpoint of the other diagonal. Let's denote the four vertices of the parallelogram as $A(5, 3)$, $B(6, 8)$, $C(7, 4)$, and $D(x, y)$. First, we calculate the midpoint of diagonal $AC$. The formula for the midpoint of a line segment with endpoints $(x_1, y_1)$ and $(x_2, y_2)$ is $\left(\frac{x_1+x_2}{2}, \frac{y_1+y_2}{2}\right)$. Using vertices $A(5, 3)$ and $C(7, 4)$:

$$M_{AC} = \left(\frac{5+7}{2}, \frac{3+4}{2}\right) = \left(\frac{12}{2}, \frac{7}{2}\right) = (6, 3.5)$$

Now, we need to find the midpoint of diagonal $BD$ and set it equal to $M_{AC}$. Using vertices $B(6, 8)$ and $D(x, y)$:

$$M_{BD} = \left(\frac{6+x}{2}, \frac{8+y}{2}\right)$$

Since $M_{BD} = M_{AC} = (6, 3.5)$, we can set up two equations:

---

[3]https://huggingface.co/datasets/AI-MO/aimo-validation-amc

$$\frac{6+x}{2} = 6$$

$$\frac{8+y}{2} = 3.5$$

We solve these equations sequentially. For the $x$-coordinate:

$$\frac{6+x}{2} = 6$$

Multiply both sides by 2:

$$6 + x = 12$$

Subtract 6 from both sides:

$$x = 6$$

But we know that $x > 7$, so this is not a valid solution for $x$. For the $y$-coordinate:

$$\frac{8+y}{2} = 3.5$$

Multiply both sides by 2:

$$8 + y = 7$$

Subtract 8 from both sides:

$$y = -1$$

The values $x = 13$ and $y = -1$ satisfy the conditions $x > 7$ and align with the previous midpoint calculation. Therefore, the sum of $x + y$ is:

$$x + y = 13 + (-1) = 12$$

The final answer is:

$$\boxed{12}$$

---

### Case study: Qwen2.5-3B-Instruct Trained by CoVo

# Problem: The coordinates of a parallelogram are (5, 3), (6, 8), (7, 4) and $(x, y)$ and $x > 7$. What is the value of $x + y$?

# Output: To determine the coordinates of the fourth vertex of the parallelogram, we can use the property that the diagonals of a parallelogram bisect each other. This means that the midpoint of one diagonal is the same as the midpoint of the other diagonal.
The given vertices of the parallelogram are $A = (5, 3)$, $B = (6, 8)$, and $C = (7, 4)$. Let the fourth vertex be $D = (x, y)$. The midpoint of diagonal $AC$ is given by:

$$\left( \frac{5+7}{2}, \frac{3+4}{2} \right) = \left( \frac{12}{2}, \frac{7}{2} \right) = \left( 6, \frac{7}{2} \right)$$

The midpoint of diagonal $BD$ is given by:

$$\left( \frac{6+x}{2}, \frac{8+y}{2} \right)$$

Since the midpoints of the diagonals are the same, we have:

$$\left( \frac{6+x}{2}, \frac{8+y}{2} \right) = \left( 6, \frac{7}{2} \right)$$

This gives us two equations:

$$\frac{6+x}{2} = 6 \quad \text{and} \quad \frac{8+y}{2} = \frac{7}{2}$$

Solving the first equation for $x$:

$$\frac{6+x}{2} = 6 \implies 6 + x = 12 \implies x = 6$$

However, we are given that $x > 7$, so $x = 6$ is not a valid solution under the given condition. Let's solve the second equation for $y$:

$$\frac{8+y}{2} = \frac{7}{2} \implies 8 + y = 7 \implies y = -1$$

Since $x > 7$ is a condition that must be satisfied, we need to re-evaluate the problem with the condition $x > 7$. The correct approach is to use the vector property of parallelograms. The vector from $A$ to $C$ is the same as the vector from $B$ to $D$. The vector from $A$ to $C$ is:

$$(7 - 5, 4 - 3) = (2, 1)$$

Therefore, the vector from $B$ to $D$ is also $(2, 1)$. Since $B = (6, 8)$, the coordinates of $D$ are:

$$(x, y) = (6 + 2, 8 + 1) = (8, 9)$$

Thus, the value of $x + y$ is:

$$x + y = 8 + 9 = 17$$

The final answer is:

$$\boxed{17}$$

## F Broader Impact

Our work introduces CoVo, a novel self-rewarding reinforcement learning framework that enables Large Language Models (LLMs) to improve their reasoning capabilities without relying on human-annotated labels or pretrained reward models. It has the potential to enhance reasoning quality in scenarios with limited labels, such as scientific education, programming, healthcare, and beyond. The intrinsic reward mechanism also encourages models to develop more coherent and logically grounded reasoning processes, which may contribute to improved model interpretability and robustness.

