# OpenReview forum: "Consistent Paths Lead to Truth: Self-Rewarding Reinforcement Learning for LLM Reasoning"
_NeurIPS.cc/2025/Conference — NeurIPS 2025 poster_

### Official Review · Reviewer_1Aaq · 2025-06-21

**Clarity:** 3
**Significance:** 3
**Originality:** 3
**Rating:** 5
**Confidence:** 3

**Summary:**

In this work authors propose an unsupervised reinforcement learning (RL) procedure for LLM finetuning using reasoning stability as a reward function. For each step of a reasoning trajectory $\tau_i$, authors define a distance to final answer as the average negative log likelihood (log perplexity) of that answer. Then for an entire trajectory, they define the consistency metric $Con(\tau_i)$ as the average number of steps which are `closer' to its own answer than alternatives, and a volatility metric $Vol(\tau_i)$ that measures how late in the reasoning process the model strays toward other answers. Using a combination of these metrics as well as a curiosity penalty, the authors arrive at their unsupervised reward function (CoVo). This reward function is demonstrated on average to outperform other unsupervised methods (EMPO, TTRL) and even some supervised methods (GRPO, Reinforce++) as measured in terms of accuracy on popular reasoning tasks.

**Questions:**

* How would authors explain the apparent `saturation' of accuracy across all finetuning methods in table 2? Is this a limitation of model size/ training resources?
* Would this objective have an implicit bias toward longer reasoning chains, where the model has `made up its mind' but continues to generate reasoning steps?
* Does this finetuning affect OOD and non-reasoning tasks such as text summarization?

**Ethical Concerns:**

["NO or VERY MINOR ethics concerns only"]

**Final Justification:**

My greatest concerns with this work before rebuttal were how marginal the performance gains were as well as unintended consequences related to the reward function such as longer reasoning chains or worse OOD performance. The experiments provided by the authors in the rebuttal (pass@k accuracy table, avg. token length during training) have helped alleviate these concerns.

**Limitations:**

Yes

**Quality:**

3

**Strengths And Weaknesses:**

Strengths:
* Significance: Analyzing the internal dynamics of CoT reasoning is a very relevant line of work
* Consistent Experimental Results: The difference in correct and incorrect trajectories are well motivated in Fig. 2, and while the improvement in accuracy on individual tasks is marginal it seems consistent across all tasks.
* Some motivating theory: Proposition 5.1 provides good explanation as to why majority voting rewards can fail

Weaknesses:
* Accuracy Saturation: The fact that this method improves maybe ~0.5% on average over unsupervised baselines (Table 2), as well as the overall saturation across all methods seems to suggest a possible limit to the power of this method to discover new reasoning patterns.
* I would be interested in seeing an in depth example of how individual reasoning trajectories qualitatively change before and after training. The authors mention in the appendix that this reward encourages "more coherent and logically grounded" reasoning but I would like more evidence here.
* I appreciate the up front discussion of complexity $\mathcal{O}(NTK)$ for this method, I would like more discussion comparing this to other unsupervised methods as this seems very expensive.
* Proposition 5.2 seems like the standard ELBO derivation for variational inference, is the new insight here just that reasoning path $s$ is the latent variable?

---

> ### Author Rebuttal · Authors · 2025-07-31
>
> We greatly appreciate the reviewer for your insightful comments and positive support.
>
> ---
>
> **[W1 & Q1] Accuracy Saturation.**
>
> Thanks. TTRL’s majority-voting-based reward enables more efficient transfer of pass@k capabilities to pass@1 in the early stages of training. However, in prolonged RL training, it eventually collapses due to reward hacking (e.g., consistently outputting "0") and lacks scalability. In contrast, our method incorporates curiosity rewards and other mechanisms to encourage diverse exploration, thus avoiding the tendency to collapse into homogeneous, fixed reasoning paths like TTRL—this is illustrated in Fig. 4 of the paper. Since we report pass@1 results and, when evaluating TTRL’s metrics, we made every effort to select its best results before collapse, the improvement in pass@1 metrics in Table 2 appears marginal. We additionally supplement the pass@k results of our method and TTRL, which serve as a metric for evaluating "model reasoning potential." The results in Tab. R1 indicate that our method has a higher "capability ceiling" than TTRL.
>
> Table R1: Pass@k on AIME24.
> | pass@k | 1    | 2    | 4    | 8    | 16   | 32   | 64   | 128  | 256  |
> | -----  | ---- | ---- | ---- | ---- | ---- | ---- | ---- | ---- | ---- |
> | TTRL |6.67 |10.00|13.33|16.67|20.00|26.67|40.00|50.00|56.67|
> | CoVo |6.67 |10.00|16.67|20.00|23.33|30.00|43.33|53.33|63.33|
>
> ---
>
> **[W2] Example of how individual reasoning trajectories qualitatively change before and after training**
>
> Thanks. Below we provide an example of question "The coordinates of a parallelogram are (5, 3), (6, 8), (7, 4) and $(x, y)$ and $x > 7$. What is the value of $x + y$?".
>
> **Before Training:**
>
> To determine the coordinates of the fourth vertex of the parallelogram, we need to utilize the property that the diagonals of a parallelogram bisect each other. This means that the midpoint of one diagonal is the same as the midpoint of the other diagonal.Let's denote the four vertices of the parallelogram as \\( A(5, 3) \\), \\( B(6, 8) \\), \\( C(7, 4) \\), and \\( D(x, y) \\).First, we calculate the midpoint of diagonal \\( AC \\).The formula for the midpoint of a line segment with endpoints \\((x_1, y_1)\\) and \\((x_2, y_2)\\) is \\(\\left( \\frac{x_1 + x_2}{2}, \\frac{y_1 + y_2}{2} \\right)\\).Using vertices \\( A(5, 3) \\) and \\( C(7, 4) \\):\\[M_{AC} = \\left( \\frac{5 + 7}{2}, \\frac{3 + 4}{2} \\right) = \\left( \\frac{12}{2}, \\frac{7}{2} \\right) = \\left( 6, 3.5 \\right)\\]Now, we need to find the midpoint of diagonal \\( BD \\) and set it equal to \\( M_{AC} \\). Using vertices \\( B(6, 8) \\) and \\( D(x, y) \\):\\[M_{BD} = \\left( \\frac{6 + x}{2}, \\frac{8 + y}{2} \\right)\\]Since \\( M_{BD} = M_{AC} = (6, 3.5) \\), we can set up two equations:\\[\\frac{6 + x}{2} = 6\\]\\[\\frac{8 + y}{2} = 3.5\\]We solve these equations sequentially.For the \\( x \\)-coordinate:\\[\\frac{6 + x}{2} = 6\\]Multiply both sides by 2:\\[6 + x = 12\\]Subtract 6 from both sides:\\[x = 6\\]But we know that \\( x > 7 \\), so this is not a valid solution for \\( x \\).For the \\( y \\)-coordinate:\\[\\frac{8 + y}{2} = 3.5\\]Multiply both sides by 2:\\[8 + y = 7\\]Subtract 8 from both sides:\\[y = -1\\]The values \\( x = 13 \\) and \\( y = -1 \\) satisfy the conditions \\( x > 7 \\) and align with the previous midpoint calculation.Therefore, the sum of \\( x + y \\) is:\\[x + y = 13 + (-1) = 12\\]
>
> **After Training:**
>
> To determine the coordinates of the fourth vertex of the parallelogram, we can use the property that the diagonals of a parallelogram bisect each other. This means that the midpoint of one diagonal is the same as the midpoint of the other diagonal.\\
> The given vertices of the parallelogram are \\(A = (5, 3)\\), \\(B = (6, 8)\\), and \\(C = (7, 4)\\). Let the fourth vertex be \\(D = (x, y)\\). The midpoint of diagonal \\(AC\\) is given by:\\[\left( \frac{5+7}{2}, \frac{3+4}{2} \right) = \left( \frac{12}{2}, \frac{7}{2} \right) = \left( 6, \frac{7}{2} \right)\\]The midpoint of diagonal \\(BD\\) is given by:\\[\left( \frac{6+x}{2}, \frac{8+y}{2} \right)\\]Since the midpoints of the diagonals are the same, we have:\\[\left( \frac{6+x}{2}, \frac{8+y}{2} \right) = \left( 6, \frac{7}{2} \right)\\]This gives us two equations:\\[\frac{6+x}{2} = 6 \quad \text{and} \quad \frac{8+y}{2} = \frac{7}{2}\\]Solving the first equation for \\(x\\):\\[\frac{6+x}{2} = 6 \implies 6 + x = 12 \implies x = 6\\]However, we are given that \\(x > 7\\), so \\(x = 6\\) is not a valid solution under the given condition. Let's solve the second equation for \\(y\\):\\[\frac{8+y}{2} = \frac{7}{2} \implies 8 + y = 7 \implies y = -1\\]Since \\(x > 7\\) is a condition that must be satisfied, we need to \textcolor{red}{re-evaluate the problem} with the condition \\(x > 7\\). The correct approach is to use the vector property of parallelograms. The vector from \\(A\\) to \\(C\\) is the same as the vector from \\(B\\) to \\(D\\). The vector from \\(A\\) to \\(C\\) is:\\[(7-5, 4-3) = (2, 1)\\]Therefore, the vector from \\(B\\) to \\(D\\) is also \\((2, 1)\\). Since \\(B = (6, 8)\\), the coordinates of \\(D\\) are:\\[(x, y) = (6+2, 8+1) = (8, 9)\\]Thus, the value of \\(x + y\\) is:\\[x + y = 8 + 9 = 17\\]The final answer is:\\[\boxed{17}\\]
>
> ---
>
> **[W3] More discussion comparing complexity to other unsupervised methods.**
>
> Thanks for the concern. We list the average cost of time in each stage within a training step in Tab. R2. Since the distance matrix is computed independently and can take advantage of parallelism, the reward calculation indeed accounts for a small proportion of the time required in a training step (nearly 24.6%). In the future, it will be possible to asynchronously compute the distance matrix during sampling to further reduce the time required for distance matrix computation.
>
> Table R2.
> |Method| Sampling Time (s)|Reward Calculation Time (s)|Training Time (s)|Reward calculation time ratio|
> | ---- | ----------------- | ---------------- | ----------------- | ----------------------------- |
> | Ground Truth Reward| 63 | 2 |87|1.3%|
> | Majority Voting Reward | 63 | 4 | 87 | 2.6% |
> | CoVo (Ours)            | 63 | 49 |87|24.6%|
>
> ---
>
> **[W4] Insight of Proposition 5.2.**
>
> While the mathematical form resembles standard ELBO, the innovation is in framing reasoning trajectories as latent variables that must be aligned with final answers—a critical adaptation for LLM reasoning tasks, where the path to an answer is often as important as the answer itself. This justifies the paper’s focus on intermediate states for reward design (CoVo’s core mechanism).
>
> ---
>
> **[Q2] Would this objective have an implicit bias toward longer reasoning chains, where the model has `made up its mind' but continues to generate reasoning steps?**
>
> Thanks for the insightful question! The objective proposed in the paper will not inherently bias toward longer reasoning chains where the model "fills in" redundant steps after settling on an answer.
>
> CoVo’s core rewards are defined based on the **consistency between intermediate reasoning states and the final answer**, not the total number of steps (length of the chain).
>
> - **Consistency** measures how often intermediate states stay "close" to the final answer (via model likelihood). A longer chain with redundant steps that still align with the final answer might maintain high consistency, but this is not rewarded *because* it is longer—it is rewarded because it remains coherent.
> -
> - **Volatility** measures how much intermediate states deviate from the final answer. A longer chain with erratic deviations (e.g., the model contradicts itself before settling on an answer) would have high volatility and be penalized, regardless of length.
>
> Thus, the reward signal incentivizes *stable, answer-aligned reasoning* but is agnostic to the chain’s length itself. A short, consistent chain and a long, consistent chain would both be rewarded, while a long, inconsistent chain would not.
>
> ---
>
> **[Q3] Does this finetuning affect OOD and non-reasoning tasks such as text summarization?**
>
> Thanks for your question! We evaluate the performance of CoVo on OOD reasoning tasks like MMLU-Pro, GPQA and CommonsenseQA in Tab. 2 of our manuscript. Additionally, we provide the performance on text summarization task in Tab. R3. The results indicate this finetuning does not affect OOD and non-reasoning tasks.
>
> Table R3.
> |               | F1 scores | ROUGE-1 |
> | ------------- | --------- | ------- |
> | CNN/DailyMail |87.7 (Origin) / 87.5 (Trained) | 28.6 (Origin) / 28.8 (Trained) |

---

> > ### Comment · Reviewer_1Aaq · 2025-08-09
> > **Response to Rebuttal**
> >
> > I'd like to thank the authors for their thorough responses to my questions and concerns.
> >
> > * The additional Pass@k accuracy comparison and qualitative example I think help to support the usefulness of this method.
> > * The cost comparison table is useful to include in the final revision as well for full transparency.
> > * Thank you for the discussion and additional experiments regarding impacts of the reward on length of reasoning chains. The lengths reported during training seem a bit noisy but I agree there doesn't seem to be much change.
> > * I appreciate the additional experiment on the OOD text summarization task
> >
> > Overall I am satisfied with the additional context provided by the authors and will advocate for this work.

---

> > > ### Author Response · Authors · 2025-08-09
> > > **Thanks for the positive support!**
> > >
> > > Thanks for the positive support and valuable comments!
> > >
> > > For the minor noise in the average response lengths, it arises because the batch of problems used for each training step is randomly sampled and therefore differs from step to step. Consequently, the variations in training samples across different steps (e.g., the sampled problems are not identical at step 50 versus step 150) naturally lead to fluctuations in the averaged response lengths. We will add the length evolving curves in the revised version to present these changes more intuitively.
> > >
> > > Once again, we sincerely appreciate your helpful suggestions. We will carefully revise our manuscript according to our discussion to provide a clear analysis of the pass@k performance and the relevant computational resources. Wishing you a wonderful day :)

---

> ### Author Response · Authors · 2025-08-01
> **Additional Discussion on Q1, Q2**
>
> **[Q1] The apparent saturation of accuracy across all finetuning methods**
>
> Recent studies [1, 2] suggest that RL may not fundamentally expand the inherent reasoning abilities of LLMs, but rather builds upon the capacity already present in the initial model. RL with verifiable rewards (RLVR) is primarily effective at converting Pass@k capabilities to Pass@1, with the model’s ultimate performance ceiling constrained by its underlying abilities. In particular, RL methods that utilize majority-vote-based rewards (e.g., TTRL) are is inherently limited by the Maj@k capacity before training. Therefore, when employing RLVR or self-rewarding RL frameworks such as CoVo, the diversity of samples during training becomes critical. Sampling diversity determines how effectively these existing reasoning capabilities can be elicited and guided from the initial model to improve Pass@1.
>
> All baseline RL methods compared in our paper are applied to instruction-tuned models, which is trained using self-generated trajectories at a high temperature during online sampling. We believe that the observed performance saturation arises because instruction-tuned models have a more restricted exploration space than base models, leading to less efficient conversion from Pass@K to Pass@1. While more sophisticated data curation strategies and training methodologies such as DAPO[3] would undoubtedly further improve RL performance, they are not the main focus of our work.
>
> -----
> **[Q2] Would this objective have an implicit bias toward longer reasoning chains**
>
> In previous rebuttal, we give analysis on why our target is independent of sequence length. To further substantiate this point, we provide additional results on the evolution of output length during the training process of Qwen2.5-3B-Instruct below. The results further support our claim that the objective does not impart an implicit bias toward the generation of longer reasoning chains.
>
> | Training Step           | 0    | 50   | 100  | 150  | 200  | 250  | 300  | 350  | 400  |
> | ----------------------- | ---- | ---- | ---- | ---- | ---- | ---- | ---- | ---- | ---- |
> | Average Response Length (Token) | 733  | 805  | 766  | 786  | 815  | 868  | 781  | 871  | 799  |
>
> -----
>
> Thanks again for the insightful questions! We will add these discussions in the revision.
>
> **References**
>
> [1] Does reinforcement learning really incentivize reasoning capacity in llms beyond the base model?
>
> [2] Mind the gap: Examining the self-improvement capabilities of large language models.
>
> [3] DAPO: An Open-Source LLM Reinforcement Learning System at Scale

---

### Official Review · Reviewer_Ejyg · 2025-06-25

**Clarity:** 3
**Significance:** 3
**Originality:** 3
**Rating:** 4
**Confidence:** 4

**Summary:**

This paper proposes CoVo, a self-rewarding RL framework for LLM reasoning that leverages internal response trajectories rather than relying on external rewards or final answer matching. By modeling reasoning paths through consistency and volatility metrics, and encoding them in a vector space, the approach enables trajectory-level reward signals. A curiosity bonus is added to encourage exploration. Empirical results demonstrate that CoVo outperforms prior self-rewarding methods and sometimes even supervised RL.

**Questions:**

1. Can the authors explain why likelihood distance was chosen over other possible measures (e.g., mutual information) for consistency/volatility scoring?

2. How does CoVo handle failure cases where wrong answers have consistent but misleading trajectories?

**Ethical Concerns:**

["NO or VERY MINOR ethics concerns only"]

**Final Justification:**

I keep my original positive score.

**Limitations:**

While the authors claim no significant limitations, we believe the method may still suffer from issues related to exploration, dependency on initial policy quality, and distance metric simplicity, as discussed above.

**Quality:**

4

**Strengths And Weaknesses:**

### Strengths:

- The paper addresses a key limitation of RL for LLMs: reliance on expensive human labels or biased reward models.
- It introduces a novel and intuitive intrinsic reward scheme based on consistency and volatility over intermediate reasoning steps.
- The proposed vector-space representation enables stable learning and avoids reward hacking.
- Results are strong across multiple reasoning benchmarks, including math and general QA.

### Weaknesses:

- Limited exploration: Although a curiosity bonus is introduced, the method still relies solely on self-generated samples without external perturbation or guided sampling, which may restrict exploration and cause convergence to locally optimal but incorrect reasoning patterns.
- Dependence on initial policy: The effectiveness of the reward signals (consistency and volatility) hinges on the quality of the initial model. Poor base models may produce unreliable reasoning paths, reducing the robustness of self-rewarding learning.
- Risk of reinforcing faulty reasoning: The framework may unintentionally reward trajectories that are confidently wrong but internally consistent, especially when no ground-truth supervision or reward model is used to validate answer correctness.
- Simplistic distance metric: The reliance on token-level negative log-likelihood as the sole measure of distance may overlook deeper semantic or structural alignment between reasoning states and final answers, potentially limiting reward fidelity in complex tasks.

---

> ### Author Rebuttal · Authors · 2025-07-31
>
> We greatly appreciate the reviewer for your insightful comments and positive support.
>
> ---
>
> **[W1] Limited exploration in RL.**
>
> Thanks for your insightful comment! We investigate the current literature and find most of works rely on self-generated samples under high temperature without external perturbation. RFTT[1] employs guided sampling via tree search, while Cheng et al.[2] augment the advantage function with an entropy-based term to promote exploration. In our work, we make an early attempt to introduce a curiosity-based reward term to encourage sampling diversity in RL, which may also be interpreted as an external perturbation. We agree that enhancing exploration during RL sampling is a highly promising direction, and we consider this an important avenue for future research.
>
> ---
>
> **[W2] Dependence on initial policy.**
>
> Thanks for your question! This concern aligns with a widely acknowledged characteristic inherent to reinforcement learning (RL) in the community[2,3]: The performance of RL-based method is inherently tied to the competence of the base policy. When the initial model exhibits fundamental limitations in task understanding or reasoning capacity, suboptimal outcomes are universally observed across RL methodologies, not uniquely in CoVo.
>
> This underscores a broader consensus in the field: the foundational capabilities of the base model serve as a critical prerequisite for effective RL fine-tuning. Advances in pre-training techniques have consistently demonstrated that stronger base models provide more robust starting points for subsequent RL optimization, a principle CoVo explicitly embraces.
>
> To validate the robustness of our framework against variations in initial policy quality, we conducted extensive experiments across model scales ranging from 3B to 7B parameters. Results show that CoVo yields consistent performance gains across all tested base models, including those with lower initial competence. This empirical evidence indicates that while CoVo, like all RL systems, benefits from stronger base policies, it maintains effectiveness and generalizability even when starting from less optimal initial states.
>
> ---
>
> **[W3] Risk of reinforcing faulty reasoning.**
>
> Thanks for your comment! This limitation is indeed inherent to unsupervised approaches. Current self-rewarding works like TTRL utilize majority voting or answer probabilities for reward assignment, where models may learn to exploit superficial shortcuts that yield unreliable high rewards (e.g., always outputting “0” for all problems can deceive majority voting into rewarding it). Our training dynamics in Fig. 4 of our manuscript demonstrate that CoVo is more effective at mitigating reward hacking and maintains long-term robustness compared to current approaches.
>
> ---
>
> **[W4 & Q1] Simplistic distance metric & other possible measures (e.g., mutual information).**
>
> Thanks for the valuable suggestion! We use the MI to replace our distance and then calculate the Pearson Correlation Coefficient to see whether there is an association between the two measures. The results in Table R1 indicate that the two metrics exhibit a high degree of similarity. Therefore, for simplicity, we opted to use the distance metric in our analyses.
>
>
> Table R1.
> |      | MATH500 | GPQA | MMLU |
> | ---- | ------- | ---- | ---- |
> |Pearson Correlation Coefficient|0.92|0.86|0.93|
>
>
>
> The reviewer’s suggestion to employ MI is valuable, and we see mutual verification that using MI and distance metrics as a promising direction for our future work, which may further improve the robustness of reward assignment.
>
> ---
>
> **[Q2] How does CoVo handle failure cases where wrong answers have consistent but misleading trajectories?**
>
> Thanks for the question! In our fully unsupervised setting, we do not have access to ground truth labels—there is no external signal to define whether an answer is "wrong" or a trajectory is "misleading." Under this design, our framework will not explicitly handle such failure cases in the sense of identifying and penalizing "wrong but consistent" trajectories, as there is no objective standard to label them as "wrong."
>
> CoVo’s success highlights that intrinsic and curiosity reward signals, derived purely from the intrinsic properties of the intermediate states in the generated trajectories, can be surprisingly potent for this elicitation process. In a well-pretrained model, answers that are consistent with the intermediate process are more likely to align with correct and coherent reasoning. CoVo leverages this by incentivizing the model to favor such consistent outputs, effectively guiding it to refine its selection from its collection of existing policy distributions without requiring external validation of correctness.
>
> ---
>
> [1] Reasoning with Reinforced Functional Token Tuning.
>
> [2] Does Reinforcement Learning Really Incentivize Reasoning Capacity in LLMs Beyond the Base Model?
>
> [3] Cognitive Behaviors that Enable Self-Improving Reasoners, or, Four Habits of Highly Effective STaRs.

---

> > ### Comment · Reviewer_Ejyg · 2025-08-05
> >
> > Thanks for the reponses! I will keep the positive score. Gook luck!

---

> > > ### Author Response · Authors · 2025-08-06
> > > **Thanks for the positive support!**
> > >
> > > We are glad that the reviewer appreciates our attempt, and sincerely thank you for the positive support! Have a nice day :)

---

### Official Review · Reviewer_QTLK · 2025-07-02

**Clarity:** 2
**Significance:** 1
**Originality:** 2
**Rating:** 3
**Confidence:** 5

**Summary:**

This paper introduces CoVo (Consistency and Volatility), a self-rewarding reinforcement learning framework for enhancing LLM reasoning capabilities without external supervision. The key insight is that correct reasoning trajectories exhibit distinct patterns: high consistency (intermediate states lead toward their final answer) and low volatility (minimal deviation toward alternative answers). The authors propose an intrinsic reward mechanism that combines these trajectory-level features through robust vector-space aggregation, supplemented by a curiosity bonus for exploration. Experiments on mathematical and commonsense reasoning benchmarks show that CoVo achieves performance comparable to supervised RL methods, improving Qwen2.5-3B-Instruct from 63.6% to 68.2% on MATH dataset.

**Questions:**

- If this method underperforms the simpler methods like TTRL, why do we need a more complex method that’s harder to implement?

**Ethical Concerns:**

["NO or VERY MINOR ethics concerns only"]

**Final Justification:**

The authors made a clarification on the data contamination, so I updated my rating a bit. However, according to real-world practice on large-scale model training, this line of work will finally lead to reward hacking. As discussed in the rebuttal stage, the reviewer does think this line of work is meaningful, but in a very different narrative.

**Limitations:**

See weakness.

**Quality:**

2

**Strengths And Weaknesses:**

- The biggest concern: Despite including tests on other domains, the training setup is restricted to Qwen on math. Therefore, I am not able to assess if the improvements come from the algorithm, or simply from the base model, as pointed out by recent work [1,2] rethinking the effectiveness of unsupervised/weakly-supervised RL methods. The gain may simply come from eliciting certain meta-capabilities [3] and then using these general reasoning skills to solve problems, both in-domain and out-of-domain, or may just learn the format.
- The improvements are very marginal compared to baselines. Specifically, compared to TTRL, the difference is usually (or always) smaller than 1.0%, and it even underperforms in many cases. RL is known to be sensitive, and so is the evaluation. In practice, even an update on training infra or python library can easily lead to a 2% accuracy change on MATH. This is the second aspect why I am not able to assess the effectiveness of the algorithm proposed in this paper.
- For the analysis in section 3.2.1, the paper says “for correct answers, the reasoning paths rapidly approach to the final answer within a few steps. In contrast, incorrect responses typically exhibit delayed convergence, with intermediate states remaining relatively distant from the final answer for an extended period.” However, the analysis does not ablate difficulty. That is, easier problems are more likely to get correct, and solving them requires fewer steps. A more rigorous analysis is required to set up such a causal correlation.

[1] Spurious Rewards: Rethinking Training Signals in RLVR. Shao et al. 2025.

[2] The Unreasonable Effectiveness of Entropy Minimization in LLM Reasoning. Agarwal et al. 2025.

[3] Cognitive Behaviors that Enable Self-Improving Reasoners, or, Four Habits of Highly Effective STaRs. Gandhi et al. 2025.

---

> ### Author Rebuttal · Authors · 2025-07-31
>
> We sincerely thank the reviewer for the constructive comments.
>
> ---
>
> **[W1]  the training setup is restricted to Qwen on math & effectiveness of our methods.**
>
> Thanks for your comments. We would like to clarify some misunderstandings regarding our settings and method.
>
> (1) First, our training setup is **not only restricted to Qwen, but also includes Llama**. The results in Tab. 2 of our original manuscript indicate that CoVo applies to different model backbones and can generalize to **general reasoning domains beyond math like MMLU, GPQA and CommonsenseQA**.
>
> (2) Second, we kindly point out that some perspectives and settings in Shao et al. (2025) and Agarwal et al. (2025) you mentioned are in fact inapplicable to our work, and their viewpoints might contain inherent flaws.
> Shao et al.’s work was conducted on Qwen2.5-7B (a pretrained base model), where using incorrect rewards for reinforcement learning (RL) could still improve the performance of the base model. Agarwal et al.’s work was also based on Qwen2.5-7B. Since base models exhibit poor instruction-following capabilities and struggle to present final answers in the desired format, the practice in their work of assigning high rewards to incorrect answers implicitly encourages the model to output in the desired format. Thus, the improvements observed in their Qwen base model might stem from eliciting certain meta-capabilities or merely learning the format. **This is precisely why all our experiments exclusively use instruct models that minimize the influence of such factors as much as possible**.
>
> Tab. R1 supplements the proportion of outputs in the desired format for both the initial Qwen2.5-7B-instruct model and the post-training model on the MATH500 dataset, showing almost no difference between the two. This indicates that our improvements do not result from learning the format. Furthermore, we conducted RL training on the Qwen2.5-7B-Instruct model following the setting of fake rewards used by Shao et al., with results presented in Table R2. It can be observed that the model’s performance improves slightly initially but completely collapses after 200 training steps, a phenomenon that does not occur with our method or ground truth rewards. The experiments in Tables R1 and R2 demonstrate that the improvements indeed stem from our method.
>
> Table R1.
> |Settings|Correct Format Rate|
> |-|-|
> |Zero-shot CoT| 99.8%|
> |Trained by CoVo| 99.6%|
>
> Table R2.
> |Training Steps| 0    | 50   | 100  | 150  | 200  | 250  | 300  | 350  | 400  | 450  |
> |-| ---- | ---- | ---- | ---- | ---- | ---- | ---- | ---- | ---- | ---- |
> | Correct Reward Eval on MATH500|63.6|62.2|64.2|65.0|65.0|65.2|65.0|66.2|66.8|67.4|
> | Incorrect Reward Eval on MATH500 |63.6|64.4|64.6|60.8|46.8|10.2|2.4|0.2|1.4|0.8|
>
> ---
>
> **[W2\&Q1] The improvements are very marginal compared to TTRL. If this method underperforms the simpler methods like TTRL, why do we need a more complex method that’s harder to implement?**
>
> Thanks! We are happy to clarify the effectiveness and potential advantages compared to concurrent work TTRL. Our method is more robust and has greater potential than TTRL.
>
> First, TTRL’s majority-voting-based reward enables more efficient transfer of pass@k capabilities to pass@1 in the early stages of training. However, in prolonged RL training, it eventually collapses due to reward hacking (e.g., consistently outputting "0") and lacks scalability. In contrast, our method incorporates curiosity rewards and other mechanisms to encourage diverse exploration, thus avoiding the tendency to collapse into homogeneous, fixed reasoning paths like TTRL—this is illustrated in Fig. 4 of the paper. Since we report pass@1 results and, when evaluating TTRL’s metrics, we made every effort to select its best results before collapse, the improvement in pass@1 metrics in Table 2 appears marginal. We additionally supplement the pass@k results on AIME24 of our method and TTRL, which serve as a metric for evaluating "model reasoning potential." The results in Table R3 indicate that our method has a higher "capability ceiling" than TTRL.
>
> Table R3: Pass@k on AIME24.
> | pass@k | 1    | 2    | 4    | 8    | 16   | 32   | 64   | 128  | 256  |
> | -----  | ---- | ---- | ---- | ---- | ---- | ---- | ---- | ---- | ---- |
> | TTRL |6.67 |10.00|13.33|16.67|20.00|26.67|40.00|50.00|56.67|
> | CoVo |6.67 |10.00|16.67|20.00|23.33|30.00|43.33|53.33|63.33|
>
> ---
>
> **[W3] the analysis does not ablate difficulty.**
>
> Thanks for your suggestions! We provide additional results in Tab. R4 and found this property is independent of task difficulty.
>
> Table R4.
> | | MATH Level 1 | MATH Level5 |
> | ----------------------- | ----------- | ----------- |
> | Consistency (Correct)   | 0.843±0.201 | 0.826±0.187 |
> | Consistency (Incorrect) | 0.225±0.142 | 0.208±0.126 |
> | Volatility (Correct)    | 0.264±0.233 | 0.274±0.241 |
> | Volatility (Incorrect)  | 0.831±0.124 | 0.825±0.113 |
>
>
> ---
>
> Thanks again for the insightful questions! We hope our responses can address your concerns. Looking forward to your reevaluation on the effectiveness and contributions of our work!

---

> > ### Comment · Reviewer_QTLK · 2025-08-05
> >
> > Thanks authors for the detailed response. However, I wonder if there is empirical evidence to ground the following claim? It seems contradictory to existing work [1] which shows Qwen-base models are already good at formated output. Also, I think it is not simply about format learning, as recent work implies issues on pattern activation [2] or even contamination [3].
> >
> > > "Since base models exhibit poor instruction-following capabilities and struggle to present final answers in the desired format, the practice in their work of assigning high rewards to incorrect answers implicitly encourages the model to output in the desired format. Thus, the improvements observed in their Qwen base model might stem from eliciting certain meta-capabilities or merely learning the format."
> >
> > Nevertheless, I appreciate authors' response on other weaknesses.
> >
> > [1] Understanding R1-Zero-Like Training: A Critical Perspective. Liu et al.
> >
> > [2] Echo Chamber: RL Post-training Amplifies Behaviors Learned in Pretraining. Zhao et al.
> >
> > [3] Reasoning or Memorization? Unreliable Results of Reinforcement Learning Due to Data Contamination. Wu et al.

---

> ### Author Response · Authors · 2025-08-05
> **Reply to the question**
>
> We appreciate the opportunity to further address the questions. The question about why RL works for **base model** is indeed an interesting discussion, with various perspectives emerging in the RL literature. We are glad to further engage in this ongoing debate; But we would like to emphasize that these considerations **do not affect the validity of our main conclusions or effectiveness**. Our main contribution is to propose a self-rewarding RL method without external supervision that yields performance comparable to supervised RL. **Importantly, we kindly emphasize that all of our experiments were conducted on the instruction-tuned model rather than the base model.**
>
> (1) **Empirical evidence.** We clarify that the claim made in our previous rebuttal refers to the setup used in Spurious Rewards [1], in which they employed templates during the training of Qwen2.5-7B (Base). However, the paper Reasoning or Memorization [2] points out that the RL gains observed for the **base model** largely reflect adaptation to the template format:
>
> > "Viewed against these baselines, the seeming 'RL gains' of Qwen-Math-7B largely reflect adaptation to the template format and merely converge to the Greedy (w/o Template) baseline, indicative of memory recall rather than genuine mathematical generalization." (Page 7)
>
> We also observe similar empirical results on Qwen2.5-3B (Base) in Tab. R4. The results indicate that the performance gain of Spurious Rewards [1] might largely stem from learning the format.
>
> Table R4.
> |Settings| Correct Format ("\\boxed") Rate on MATH500 |
> |-|-|
> |w/ template|33.2%|
> |w/o template|89.8%|
>
> Conversely, we have provided additional investigation in the previous rebuttal (Tab. R1, R2) to validate the rationality of using **instruction-tuned models**. Moreover, the Fig. 3 in Reasoning or Memorization [2] also indicates that instruction-tuned models are indeed more challenging to improve. This highlights the effectiveness and persuasiveness of our proposed method in turn.
>
> > "Additionally, we apply the same RLVR procedure to Qwen2.5-Math-7B-Instruct and discover that the resulting gains are marginal when compared with those of Qwen2.5-Math-7B, indicating that the two Qwen variants exhibit differential sensitivity to RLVR." (Page 7)
>
> (2) **Pattern activation.** We agree with the reviewer that the performance gain of base model "not simply about format learning", as stated in our previous rebuttal "the improvements observed in their Qwen base model might stem from **eliciting certain meta-capabilities**  or merely learning the format".
> Recent studies [3, 4] suggest that RL may not fundamentally expand the inherent reasoning abilities of LLMs, but rather elicit the capacity already present in the initial model. RLVR is primarily effective at converting Pass@k capabilities to Pass@1, with the model’s ultimate performance ceiling constrained by its underlying abilities. **CoVo achieves a higher Pass@k score than TTRL** (Tab. R3 in previous rebuttal) as k increases, indicating that our approach not only improves Pass@1 accuracy, but also more effectively preserves the capabilities of the initial model.
>
> There remains considerable ongoing discussion within the community regarding whether the performance gains achieved by RL methods (both supervised and unsupervised) are attributable to certain meta-capabilities or existing patterns in the initial model. While we acknowledge this is a promising direction for further investigation, it lies outside the primary motivation of our paper and does not affect the validity of our method. **Both supervised and unsupervised RL might simply elicit and guide these pre-existing patterns or capabilities; this, however, does not influence the empirical effectiveness demonstrated for our approach, which achieves performance comparable to supervised RL.**
>
> (3) **Data contamination on Qwen.** We agree that data contamination is a pervasive and critical concern in evaluating model performance. However, in our work, data contamination is not directly relevant to the validity of our method. We focus on longitudinal performance improvement that our method can enhance the capabilities of a given initial model itself, rather than making cross-model comparisons. Moreover, we would like to emphasize that our experiments are **not limited to Qwen models but validate our method using Llama as well**.
>
> ----
> We appreciate the opportunity to further clarify our motivation and why CoVo is effective from different perspectives.
>
> If you have further questions, please feel free to reach out. Looking forward to your reevaluation of our work!
>
> ----
> References
>
> [1] Spurious Rewards: Rethinking Training Signals in RLVR.
>
> [2] Reasoning or Memorization? Unreliable Results of Reinforcement Learning Due to Data Contamination.
>
> [3] Does Reinforcement Learning Really Incentivize Reasoning Capacity in LLMs Beyond the Base Model?
>
> [4] Echo Chamber: RL Post-training Amplifies Behaviors Learned in Pretraining.

---

> > ### Comment · Reviewer_QTLK · 2025-08-06
> >
> > Thanks for the response. However, I do think this affects the conclusion drawn in this paper, and the use of the instruct model makes the contamination even more suspicious, as it is even more likely that the instruct model has been trained on open-source datasets used in this paper. To be fair, I do think the unsupervised-RL line of work is meaningful, in that it reminds people to appreciate the effectiveness of base models, rather than considering this as a scalable RL method in practice.

---

> > > ### Author Response · Authors · 2025-08-06
> > > **Reply to concerns on contamination**
> > >
> > > Thank you once again for your thoughtful and constructive review, which has significantly contributed to improving the clarity and quality of our paper. Below, we sincerely address the new comments you raised.
> > >
> > > (1) As highlighted in Reasoning or Memorization [1], using leakage-free datasets for training is preferable:
> > >
> > > > "Using these leakage-free datasets, we further show that under the RL protocol, only accurate reward signals yield steady improvements that surpass the model’s performance ceiling in mathematical reasoning, whereas noisy or incorrect rewards do not." (Page 1)
> > >
> > > As presented in our original manuscript (Appendix D), the source of our training dataset mainly consists of Numina-Math (Jul., 2024), Tulu3 MATH (Nov., 2024), and OpenR1-Math-220k (Feb., 2025). Note that Qwen2.5 series was released in Sep., 2024. Most of our training dataset was released after Qwen2.5, while most works in current RL literature are also based on math-related datasets for training. Therefore, we believe the percentage of data contamination is relatively low, making our training data a comparatively clean starting point for RL experiments.
> > >
> > > Moreover, the model collapses rapidly on incorrect rewards when training on our dataset, as shown in Tab. R2 (previous rebuttal). This aligns with the findings in Reasoning or Memorization [1], which indicates that our training data has a low risk of data leakage. This further validates the rationality of data selection for training.
> > >
> > > > "Under inverted rewards, the model collapses rapidly. These findings suggest that for problems not leaked during pretraining, only correct reward signals can effectively guide the model toward improved reasoning performance." (Page 11)
> > >
> > > (2) Second, as figured out in Reasoning or Memorization [1], the Llama model is at minimal risk of data contamination. **Notably, we achieved competitive improvements on the Llama model as well, further confirming the effectiveness of our approach.**
> > >
> > > >  "As Figure 1 illustrates, we observe that Qwen can accurately complete the problem statement and provide the correct answer, whereas Llama does not." (Page 3)
> > >
> > > Moreover, Fig. 3 in Reasoning or Memorization [2] also indicates that **instruction-tuned models are indeed more challenging to improve** than the base model. This highlights the effectiveness and persuasiveness of our proposed method.
> > >
> > > (3) The potential risk of data leakage in Qwen is a concern. However, such contamination primarily affects the model's performance on benchmark datasets in terms of "absolute performance." Our research, by contrast, focuses on achieving further improvements using the same initial model. If improvements are observed under these conditions, this demonstrates the effectiveness of our method. (P.S. We employ the instruction-tuned model to avoid any spurious improvements that might arise from instruction-following or format-learning capabilities learned by the base model.)
> > >
> > > Moreover, we report results on AIME2025 and Omni-Math datasets that were released after the Qwen2.5 series to further validate the effectiveness of our method. The results in Tab. R5 demonstrates that **our approach yields meaningful improvements even on evaluation data that were not leaked**.
> > >
> > > Table R5
> > > |Method|AIME25|Omni-Math-500|
> > > |-|-|-|
> > > |Initial|3.33|26.0|
> > > |Reinforce++|10.00|28.6|
> > > |CoVo (Ours)|10.00|28.8|
> > >
> > > (4) Lastly, we believe that self-rewarding RL methods like CoVo could prove valuable for downstream applications where human expert annotation of gold-standard answers is cost-prohibitive and unsustainable [2]. While RL techniques, including CoVo, may not instill entirely new reasoning capabilities beyond what pre-training provides, their role in significantly enhancing the sampling efficiency and eliciting those pre-trained abilities is of paramount practical importance.
> > >
> > > ---
> > >
> > > We appreciate the opportunity to clarify our settings from both model selection and training data selection to demonstrate the effectiveness of our method. If you have further questions, please feel free to reach out, and we are open to discussing.
> > >
> > > We sincerely look forward to your reevaluation of our work and would very appreciate it if you could raise your score to boost our chances of more exposure to the community. Thank you very much!
> > >
> > > ---
> > > References
> > >
> > > [1] Reasoning or Memorization? Unreliable Results of Reinforcement Learning Due to Data Contamination.
> > >
> > > [2] Right Question is Already Half the Answer: Fully Unsupervised LLM Reasoning Incentivization.

---

> ### Author Response · Authors · 2025-08-09
> **Looking forward to your feedback!**
>
> Dear Reviewer QTLK,
>
> We sincerely appreciate the reviewer for the constructive comments. As a gentle reminder, there are fewer than 4 hours remaining in the discussion period. We would appreciate it if you could confirm whether you have any concerns regarding our latest response or any additional points you would like us to address.
>
> Best regards,
>
> Authors of CoVo

---

### Official Review · Reviewer_RJ6X · 2025-07-05

**Clarity:** 3
**Significance:** 3
**Originality:** 3
**Rating:** 4
**Confidence:** 2

**Summary:**

This paper proposes CoVo, a novel self-rewarding reinforcement learning framework for improving the reasoning abilities of LLMs without relying on external supervision. The method utilizes the consistency and volatility of intermediate reasoning steps across sampled trajectories to construct intrinsic rewards, and introduces a curiosity bonus to encourage exploration. The authors conduct extensive empirical studies and theoretical analysis, showing that correct answers tend to follow consistent and low-volatility reasoning paths.

**Questions:**

1. How sensitive is CoVo to the number of sampled reasoning paths per query? Could performance degrade with fewer samples?
2. Would combining CoVo with external reward models further enhance performance, or cause interference?

**Ethical Concerns:**

["NO or VERY MINOR ethics concerns only"]

**Final Justification:**

After reading the author's rebuttal, I have decided to keep my score unchanged.

**Limitations:**

yes

**Quality:**

3

**Strengths And Weaknesses:**

Strength:

1. The paper introduces a principled and original approach for constructing self-reward signals using internal trajectory structure, addressing a key limitation in prior self-rewarding methods.
2. The authors provide a solid theoretical justification by framing the learning objective as a variational inference problem over reasoning paths, adding conceptual clarity to the method.
3. The paper is well written and easy to follow.

Weakness:
1. The computation of distance matrices and multiple trajectory sampling per prompt may lead to significant training costs.
2. Experiments are restricted to relatively small models, generalization to larger models (e.g., 13B+) or production settings is unclear.

---

> ### Author Rebuttal · Authors · 2025-07-31
>
> We greatly appreciate the reviewer for your insightful comments and positive support.
>
> ---
>
> **[W1] The computation of distance matrices and multiple trajectory sampling per prompt may lead to significant training costs.**
>
> Thanks for the insightful comments. We list the average cost of time in each stage within a training step in Tab. R1. Since the distance matrix is computed independently and can take advantage of parallelism, the reward calculation indeed accounts for a small proportion of the time required in a training step (nearly 24.6%). In the future, it will be possible to asynchronously compute the distance matrix during sampling to further reduce the time required for distance matrix computation.
>
> Table R1.
> |Method| Sampling Time (s)|Reward Calculation Time (s)|Training Time (s)|Reward calculation time ratio|
> | ---- | ----------------- | ---------------- | ----------------- | ----------------------------- |
> | Ground Truth Reward| 63 | 2 |87|1.3%|
> | Majority Voting Reward | 63 | 4 | 87 | 2.6% |
> | CoVo (Ours)            | 63 | 49 |87|24.6%|
>
> ---
>
> **[W2] Experiments are restricted to relatively small models, generalization to larger models (e.g., 13B+) or production settings is unclear.**
>
> Thanks for the helpful suggestion. In our paper, we provide model sizes of 3B & 7B. As suggested, we provide additional results on Qwen2.5-14B-Instruct to further validate the effectiveness of CoVo when model parameter scaling. The results in Tab. R2 shows meaningful improvements, which demonstrates the effectiveness of our approach.
>
> Table R2.
> ||MATH500|AMC-23|Olympiad Bench| MMLU-Pro | GPQA | CommonsenseQA |
> |-| ------- | ------ | -------------- | -------- | ---- | ------------- |
> |Initial|80.6|62.5|44.2|63.6|45.5|82.8|
> |GRPO   |84.8|65.0|46.7|64.1|46.0|83.6|
> |CoVo (Ours)|85.2|67.5|46.9|64.8|46.4|83.2|
>
> ---
>
> **[Q1] How sensitive is CoVo to the number of sampled reasoning paths per query? Could performance degrade with fewer samples?**
>
> Thanks for the valuable feedback. The number of sampled paths per query is a critical hyperparameter to RL techniques, including CoVo.
>
> (1) Reinforcement Learning (RL) optimization algorithms, such as GRPO and RLOO, necessitate sampling multiple responses for the same question. The advantage function is then estimated based on the rewards of these responses. Variations in the number of sampled responses lead to changes in the mean and standard deviation of intra-group rewards, which in turn affect the calculation of the relative advantage for each response. Specifically, if the number of samples is too small, it may result in inaccurate estimation of the advantage, thereby impairing the effectiveness of policy updates. Conversely, with a larger number of samples, intra-group samples exhibit greater diversity. This enhances the precision of the advantage function calculation, enabling the model to adjust its policy more accurately.
>
> (2) Therefore, the number of sampled paths per query is also important to CoVo, a characteristic inherited from its foundation in RL. An insufficient number of samples results in inadequate diversity among the sampled reasoning paths. This poses difficulty for the intrinsic reward to perform meaningful comparisons across distinct answers to estimate the advantage function. In practice, the number of sampled paths per query is typically set to 8, 16, 32, or 64. In our work, we opted for 16 for the balance between performance and computational resources.
>
> ---
>
> **[Q2] Would combining CoVo with external reward models further enhance performance, or cause interference?**
>
> Thanks for the constructive comments. We have provided the performance results in weakly supervised scenarios, where approximately only 20% of the data are labeled to obtain external rewards. The results in Table R3 demonstrate that our method can enhance performance when combined with external rewards.
>
> Table R3: Performance on Llama3.2-3B-Instruct with the combination of external reward.
> |Method|MATH500| AMC-23 |Olympiad Bench | MMLU-Pro | GPQA | CommonsenseQA |
> |--|-------| ------ | -------------- | -------- | ---- | ------------- |
> |CoVo (Origin)|51.2|25.0|19.6|37.2|32.1|74.4|
> |CoVo (New)|51.6|27.5|20.1|37.1|32.4|74.3|

---

> ### Comment · Reviewer_RJ6X · 2025-08-08
>
> After reading the author's rebuttal, I have decided to keep my score unchanged.

---

> > ### Author Response · Authors · 2025-08-08
> > **Thanks!**
> >
> > Thanks for your positive support and constructive comments, which help our paper to be improved. Have a nice day :)

---

### Note · Authors · 2025-08-12

Dear AC and Reviewers,

We sincerely thank the reviewer for the constructive suggestion and insightful comments. This work identifies an interesting yet underexplored insight that correct responses from LLMs often exhibit *consistent trajectory patterns*, indicating that *intermediate reasoning states* (instead of final answers as used in existing works) can serve as a reliable foundation for self-supervised reward construction. We appreciate reviewers found **analyzing the internal dynamics of CoT reasoning meaningful** (Reviewer 1Aaq), **self-reward signal is novel and principled** (Reviewer Rj6X, Ejyg), and **solid theoretical justification further clarifies our method** (Reviewer Rj6X, 1Aaq).

In the rebuttal, we have addressed key concerns as follows:

- **More Computational Costs Discussion** (Reviewer Rj6X, 1Aaq). We provide a detailed comparison of computational time per training step with other methods, and propose optimization strategies to further reduce the reward computation overhead.
- **Larger model size** (Reviewer Rj6X). We provide results on 14B model to further validate the effectiveness of CoVo when model size scaling.
- **Improvements compared to TTRL** (Reviewer QTLK, 1Aaq). We additionally provide pass@k results to demonstrate our method may have a higher "capability ceiling" than TTRL.
- **Discussions on the effectiveness of CoVo** (Reviewer QTLK). We provide additional experiments to show the improvement gain is not due to format learning or incorrect rewards. Reviewer QTLK raised a question of potential data contamination of the Qwen model itself. We further clarify that our training data is with low risk of contamination, making a clean starting point for RL experiments. Moreover, the LLaMA model is at low risk of data contamination and **we achieved competitive improvements on the LLaMA as well** in our original manuscript.

Overall, we believe that self-rewarding RL methods like CoVo could be valuable for downstream scenarios where human annotation of ground truth labels is costly and unsustainable. While RL methods, including CoVo, may not instill entirely new reasoning capabilities beyond the initial model, their role in enhancing the sampling efficiency and eliciting those pre-trained abilities is of great practical importance.

We believe our clarifications and responses have addressed the reviewers’ concerns, and we sincerely thank the AC and reviewers for your valuable time and constructive feedback.

Sincerely,

Authors

---

### Decision · Program_Chairs · 2025-09-17

**Decision:**

Accept (poster)

**Comment:**

(a) Summarize the scientific claims and findings of the paper based on your own reading and characterizations from the reviewers.

This paper introduces CoVo, a novel self-rewarding reinforcement learning framework designed to enhance the reasoning capabilities of Large Language Models (LLMs) without requiring external supervision. The central insight is that correct reasoning trajectories exhibit distinct dynamic patterns: their intermediate states tend to converge consistently toward their own final answer (high Consistency) while showing minimal deviation toward alternative answers (low Volatility). The authors propose an intrinsic reward signal, CoVo, that quantifies these two properties using model likelihoods across multiple sampled trajectories. This reward, augmented with a curiosity bonus to encourage exploration, allows the LLM to fine-tune itself. The authors also provide a theoretical justification, framing the CoVo objective as a form of variational inference where reasoning paths are treated as latent variables. Experiments on various reasoning benchmarks show that CoVo can achieve performance comparable to, and in some cases surpassing, supervised RL methods.

(b) What are the strengths of the paper?

The paper has several notable strengths, as highlighted by the reviewers:

Novelty and Significance: The paper tackles the critical problem of scaling up LLM reasoning capabilities without relying on expensive, human-annotated data. The core idea of using the internal consistency and volatility of reasoning paths as a self-reward signal is novel, intuitive, and principled (Reviewers RJ6X, Ejyg, 1Aaq).

Methodological Soundness: The proposed CoVo reward is well-motivated by empirical analysis showing clear differences between correct and incorrect reasoning trajectories. Furthermore, the authors provide a solid theoretical connection to variational inference, which adds conceptual depth to their approach (Reviewers RJ6X, 1Aaq).

Thorough Evaluation and Rebuttal: The authors conduct extensive experiments on multiple datasets and model families (Qwen and Llama), demonstrating consistent improvements. The author rebuttal was very strong, providing new experiments that effectively addressed key reviewer concerns, including results on a larger 14B parameter model, a computational cost analysis, and a more nuanced evaluation using pass@k metrics.

(c) What are the weaknesses of the paper? What might be missing in the submission?

The main weaknesses identified during the review process include:

Marginal Pass@1 Improvements: A key concern, raised by Reviewers QTLK and 1Aaq, was that the improvements in pass@1 accuracy over some unsupervised baselines (e.g., TTRL) were modest. While the authors effectively contextualized this with superior pass@k performance and robustness against training collapse, the headline gains remain small.

Computational Overhead: The proposed method requires sampling multiple trajectories and computing distance matrices, which introduces a non-trivial computational cost compared to standard fine-tuning or simpler reward schemes (Reviewers RJ6X, 1Aaq).

Dependence on Initial Policy: As with most RL-based methods, the effectiveness of CoVo is contingent on the quality of the base model. A poor initial model may not generate sufficiently coherent reasoning paths for the self-reward mechanism to be effective (Reviewer Ejyg).

(d) Provide the most important reasons for your decision to accept/reject.

I recommend accepting this paper. The primary reason is that the work presents a well-executed and novel contribution to the important and challenging area of unsupervised LLM alignment. The authors' central idea is clever and their empirical validation is solid.

Crucially, the authors' engagement during the rebuttal period was effective. They provided substantial new evidence that directly addressed the reviewers' most critical concerns. The introduction of pass@k results was a key addition, as it convincingly argued that CoVo has a higher "capability ceiling" and greater stability than the TTRL baseline, justifying its added complexity despite modest pass@1 gains. The addition of results on a 14B model and a clear cost breakdown further strengthened the paper. While one reviewer (QTLK) remained skeptical, their final concerns appeared to stem from a broader disagreement with the research direction rather than specific, unaddressed flaws in this particular manuscript. The positive assessments from the other three reviewers, especially the strong advocacy from Reviewer 1Aaq post-rebuttal, confirm that the paper makes a valuable contribution to the field.

(e) Summarize the discussion and changes during the rebuttal period.

The discussion period was active and significantly improved the paper.

Initial Reviews: The initial reviews were mixed, with three reviewers leaning positive (4, 4, 5) and one leaning negative (3). Positive reviewers praised the paper's novelty and principled approach but raised concerns about computational cost, generalization to larger models, and the magnitude of improvements. The negative reviewer (QTLK) questioned the significance of the results, citing marginal gains, potential confounders (e.g., format learning), and broader community concerns about data contamination in the base models.

Author Rebuttal: The authors provided a comprehensive rebuttal with several new experiments. Key points included:

Model Scale (for R-RJ6X): Provided new, positive results on a Qwen-14B model.

Marginal Gains (for R-QTLK, R-1Aaq): Presented pass@k results showing that while pass@1 gains over TTRL were small, CoVo achieved a much higher score at larger values of k and did not suffer from the training collapse observed in TTRL.

Confounders (for R-QTLK): Argued that using instruction-tuned models (not base models) mitigates format-learning issues and provided experiments showing their method collapses with incorrect rewards, unlike what is reported in some cited works.

Data Contamination (for R-QTLK): Defended their setup by noting their training data largely post-dates the Qwen model release and pointed to strong results on Llama models, which are considered to have lower contamination risk.

Computational Cost (for R-RJ6X, R-1Aaq): Provided a breakdown showing the reward calculation adds ~25% to the training step time.

Post-Rebuttal Discussion: The rebuttal was highly effective. Reviewer 1Aaq was fully convinced, raised their score, and became a strong advocate for the paper. Reviewer Ejyg maintained their positive stance. Reviewer RJ6X acknowledged the rebuttal but kept their score. Reviewer QTLK engaged further on the topic of data contamination but did not seem convinced by the authors' arguments, ultimately framing the work within a different "narrative" for the field.

Final Weighing: I found the authors' responses and new evidence compelling. They successfully addressed the most actionable criticisms and strengthened their claims, particularly regarding the method's stability and potential compared to simpler baselines.